# Collagen-rich omentum is a premetastatic niche for integrin α2-mediated peritoneal metastasis

Yen-Lin Huang[1], Ching-Yeu Liang[1], Danilo Ritz[2], Ricardo Coelho[3,4,5], Dedy Septiadi[6], Manuela Estermann[6], Cécile Cumin[1], Natalie Rimmer[1], Andreas Schötzau[1], Mónica Núñez López[1], André Fedier[1], Martina Konantz[7], Tatjana Vlajnic[8], Diego Calabrese[9], Claudia Lengerke[7,10], Leonor David[3,4,5], Barbara Rothen-Rutishauser[6], Francis Jacob[1]*, Viola Heinzelmann-Schwarz[1,11]*

[1]Ovarian Cancer Research, Department of Biomedicine, University Hospital Basel and University of Basel, Basel, Switzerland; [2]Proteomics core facility, Biozentrum, University of Basel, Basel, Switzerland; [3]Differentiation and Cancer group, Institute for Research and Innovation in Health (i3S), University of Porto, Porto, Portugal; [4]Institute of Molecular Pathology and Immunology of the University of Porto (IPATIMUP), Porto, Portugal; [5]Faculty of Medicine, University of Porto, Porto, Portugal; [6]Adolphe Merkle Institute, University of Fribourg, Fribourg, Switzerland; [7]Stem Cells and Hematopoiesis, Department of Biomedicine, University Hospital Basel, University of Basel, Basel, Switzerland; [8]Institute of Pathology, University Hospital Basel, Basel, Switzerland; [9]Histology Core Facility, Department of Biomedicine, University Hospital Basel and University of Basel, Basel, Switzerland; [10]Department of Internal Medicine, Internal Medicine II, Hematology, Oncology, Clinical Immunology and Rheumatology, University Hospital Tübingen, Tübingen, Germany; [11]Gynecological Cancer Center, University Hospital Basel, Basel, Switzerland

*For correspondence:
francis.jacob@unibas.ch (FJ);
viola.heinzelmann@usb.ch (VH-S)

Competing interests: The authors declare that no competing interests exist.

**Abstract** The extracellular matrix (ECM) plays critical roles in tumor progression and metastasis. However, the contribution of ECM proteins to early metastatic onset in the peritoneal cavity remains unexplored. Here, we suggest a new route of metastasis through the interaction of integrin alpha 2 (ITGA2) with collagens enriched in the tumor coinciding with poor outcome in patients with ovarian cancer. Using multiple gene-edited cell lines and patient-derived samples, we demonstrate that ITGA2 triggers cancer cell adhesion to collagen, promotes cell migration, anoikis resistance, mesothelial clearance, and peritoneal metastasis in vitro and in vivo. Mechanistically, phosphoproteomics identify an ITGA2-dependent phosphorylation of focal adhesion kinase and mitogen-activated protein kinase pathway leading to enhanced oncogenic properties. Consequently, specific inhibition of ITGA2-mediated cancer cell-collagen interaction or targeting focal adhesion signaling may present an opportunity for therapeutic intervention of metastatic spread in ovarian cancer.

## Introduction

Peritoneal metastasis is the leading cause of mortality in epithelial ovarian cancer (EOC) and results in a 17–39% 5 year survival rate due to the majority of patients being diagnosed at advanced stage of disease (>75%, FIGO stage III/IV) (*Halkia et al., 2012*; *Vaughan et al., 2011*). The accompanying development of massive ascites, a transudate predominantly seen in EOC, fosters the rapid spread

of tumor cells through transcoelomic mechanisms in the peritoneal cavity including the omentum as a preferred site (*Naora and Montell, 2005*). Ascites-derived tumor cells (ATCs) contribute to an invasive and chemo-resistant cell population with high potential to form metastasis (*Kipps et al., 2013*; *Lawrenson et al., 2011*; *Sodek et al., 2009*). Emerging studies highlight the dynamic inter-play between malignant, stromal and immune cells in the premetastatic niche promoting cancer cell adhesion, invasion, and progression leading to peritoneal metastasis. The peritoneal tumor microen-vironment consists of mesothelial cells (*Kenny et al., 2014*; *Li et al., 2019*), adipocytes (*Nieman et al., 2011*) cancer-associated fibroblasts (*Gao et al., 2019*; *Lau et al., 2017*), macro-phages (*Etzerodt et al., 2020*), as well as non-cellular components such as growth factors and the extracellular matrix (ECM) (*Cho et al., 2015*; *Gilkes et al., 2014*) which provide critical cues for sus-tained tumor growth and metastasis. Collagens are major components of the ECM and account for ~90% of the total ECM, not only providing structural integrity but also regulating diverse cellular functions (*Gilkes et al., 2014*). Dysregulation of collagens positively correlate with poor outcome in EOC patients (*Cheon et al., 2014*). Moreover, altered collagen organization also influences tissue mechanics and compromises drug delivery (*Loeffler et al., 2006*) thus contributes to the poor response to chemotherapy in EOC patients (*Jazaeri et al., 2005*). Despite solid evidence on the role of collagens in the already established tumor microenvironment, their contribution at the metastatic niche of early peritoneal dissemination remains largely unexplored.

Functional integrins act as heterodimeric cell-surface receptors that mediate cell adhesion to the ECM in tumor progression. Altered integrin expression not only supports oncogenic growth factor signaling, but also facilitates anoikis resistance of tumor cells in various malignancies (*Desgrosellier and Cheresh, 2010*; *Lu et al., 2008*). A subset of integrins, namely α1, α2, α10, and α11 exhibiting high collagen-binding affinity, are responsible for transducing biochemical or mechanical cues to regulate a variety of cellular functions such as cell proliferation, adhesion, migra-tion, and hemostasis (*Adorno-Cruz and Liu, 2019*). Recent studies highlight the critical role of integ-rin α5 (ITGA5) in mediating cancer cell interaction to fibronectin during peritoneal metastasis through an c-Met/FAK/Src-dependent signaling (*Gao et al., 2019*; *Iwanicki et al., 2011*; *Kenny et al., 2014*). However, considering the high abundance and dysregulation of collagens in the tumor microenvironment, there is a need for understanding the molecular mechanism of how collagens and collagen-binding integrins regulate tumor progression and metastasis.

In this study, we demonstrate that the collagen-binding integrin α2 (ITGA2) on EOC cells is required for direct and selective cell adhesion to various collagens which are enriched in the omen-tum. Using multiple cell lines and patient-derived samples, we provide functional evidence that ITGA2 promotes cancer cell adhesion, anoikis resistance, and tumor spheroid-mediated mesothelial clearance in vitro as well as peritoneal metastasis in vivo. (Phospho-)proteomics reveal that ITGA2-dependent signaling is transduced by phosphorylation of focal adhesion kinase and mitogen-acti-vated protein kinase (MAPK) signaling pathway. Furthermore, functional blockade targeting ITGA2-collagen mediated cell adhesion or inhibiting downstream focal adhesion kinase activity provides the basis of the promising therapeutic potential for intervening with peritoneal metastasis.

## Results

### Collagen-rich omentum is a metastatic niche for ITGA2-mediated cancer cell adhesion

To investigate the ECM composition of the omentum, representing the preferred metastatic site of EOC, we performed proteomic analysis of normal omentum from benign diseases and omental metastases from advanced EOC. We identified total 1298 proteins significantly upregulated in omental metastases including several ECM-related proteins dominant by collagens (e.g. COL1A1, COL3A1, and COL5A1), collagen-modifying hydroxylase (PLOD1 and PLOD2), and collagen-degrad-ing matrix metalloproteinase (MMP2 and MMP9) (*Figure 1A* and *Supplementary file 1a*). Impor-tantly, *COL1A1*, *COL3A1*, and *COL5A1* have been correlated with disease progression (*Gilkes et al., 2014*) and are predictors of overall survival for EOC patients as shown in the pooled hazard ratio model obtained from *CuratedOvarianData* (HR 1.16–1.20, n = 2970) (*Figure 1B*). In line with a previous study showing that extensive collagen deposition is recognized as a pathological characteristic resulting in increasing tumor stiffness and promoting metastasis of EOC (*Pearce et al.,*

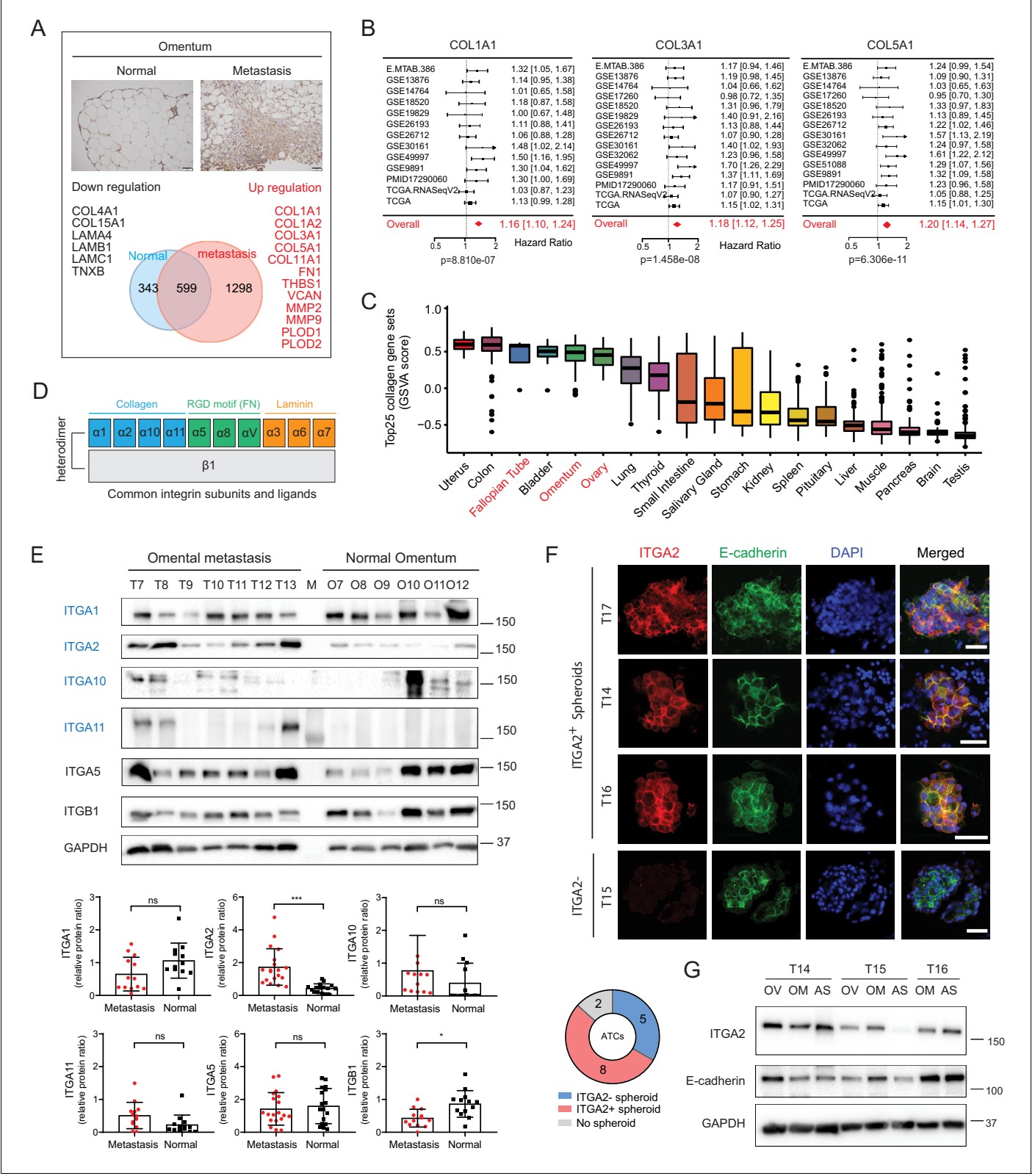

**Figure 1.** Altered collagen expression predicts poor outcome in EOC patients coinciding with ITGA2 expression. (**A**) Proteomic analysis identifies up- and downregulated ECM-associated proteins in omental metastasis *versus* normal omentum tissue (n = 8). Representative immunohistochemical staining of normal and metastatic omentum for COL1A1. Scale bar 50 μm. (**B**) Forest plots of the expression of collagens (COL1A1, COL3A1, and COL5A1) as univariate predictors of overall survival, using the *curatedOvarianData* (n = 2970) applicable expression and survival information. Hazard

Figure 1 continued

ratio (HR) significantly larger than one indicates positive correlation to poor outcome in EOC patients. (C) Box-whisker plots of top 25 collagens gene set variation analysis (GSVA) in 20 non-diseased tissues from GTEx RNA-seq dataset. (D) A schematic figure of integrin receptors and their corresponding ECM ligands. (E) Representative western blot shows the expression of collagen-binding integrins α1, α2, α10, α11, as well as integrin α5 and β1 in omental metastasis and normal omentum. Bar charts with relative integrin expression as mean ± SD (**p<0.05, ***p<0.001; n = 13–19). (F) Representative immunofluorescence images with membranous E-cadherin (green) and ITGA2 (red) staining in ATCs. Pie chart summarizes percentage of ITGA2+ tumor spheroids from EOC patients (n = 15). Scale bar 50 μm. (G) Western blot analysis of ITGA2 and E-cadherin expression in matched EOC patient samples, ascites (AS), primary (OV) and omental metastasis (OM).

The online version of this article includes the following figure supplement(s) for figure 1:

**Figure supplement 1.** Collagens and ECM-associated gene expression correlates with poor survival in patients with ovarian cancer.
**Figure supplement 2.** Hierarchical cluster analysis of collagen-expression genes in EOC and fibroblast cell lines.
**Figure supplement 3.** ITGA2 expression correlates with collagen expression in TCGA and GTEx dataset.

---

*2018*). Although collagens have been widely accepted to elicit biochemical or biophysical signaling in tumor progression in the established tumor microenvironment (*Xu et al., 2019*), the interplay between collagens and tumor cells in the premetastatic niche remains unclear. Here, we identified that collagen-encoding genes share a similar mRNA expression profile among normal omentum, ovary and fallopian tube using the Gene Set Variation Analysis (GSVA) of the Genotype-Tissue Expression (GTEx) dataset (n = 17,382) (*Figure 1C*). In particular, type I collagen (encoded by *COL1A1* and *COL1A2*) shows the highest collagen gene expression in normal ovary and omentum as well as in the omental metastases (*Figure 1—figure supplement 1A–B*). It is also noticed that the origins of high *COL1A1*, *COL1A2*, and *COL3A1* expression positively correlate with fibroblast-specific markers (FAP) and smooth muscle actin (ACTA2) (*Figure 1—figure supplement 1C*), whereas *COL4A1*, *COL4A2,* and *COL18A1* seem to be predominantly expressed in ovarian cancer cells (*Figure 1—figure supplement 2*). Together, this finding prompted us to investigate the role of collagen as a potential chemotactic matrix protein and to elucidate the interplay between collagen and associated receptors (integrins) in initial cancer cell adhesion to the omentum.

By assessing the tissue expression of major collagen-binding integrins (ITGA1, ITGA2, ITGA10, and ITGA11, *Figure 1D*), ITGA10 and ITGA11 revealed limited expression whereas ITGA1 showed a ubiquitous expression pattern in both, normal tissues and omental metastases (*Figure 1E*, *Figure 1—figure supplement 3A–B*). Of note, among the investigated integrin subunits, only ITGA2 is significantly upregulated in omental metastases (*Figure 1E*). Moreover, elevated *ITGA2* expression positively correlates with the defined collagen gene set in The Cancer Genome Atlas (TCGA) (R = 0.32, p=1.2e-10) (*Figure 1—figure supplement 3C–D*) and GTEx (R = 0.34, p=1.9e-09) dataset (*Figure 1—figure supplement 3E*) suggesting a potential interaction between ITGA2$^+$ tumor cells and enriched collagens in the premetastatic niche. Detachment of EOC cells from primary tumor and the accumulation of ascites is widely accepted as a common route of peritoneal metastasis (*Yeung et al., 2015*). Therefore, we examined whether ascites-derived tumor cells (ATCs) as well as matched primary (OV) and metastatic tissues (OM) express ITGA2, and identified that 61.5% of patients-derived ATCs are E-cadherin and ITGA2 positive. This finding suggests a potential advantage of elevated ITGA2$^+$ tumor cells in promoting peritoneal dissemination to collagen-rich omentum (*Figure 1F-G*).

## Establishment and characterization of an ΔITGA2 ovarian cancer cell line model

To study the specific collagen-binding integrins in cancer cell dissemination, we analyzed the transcriptome of the Cancer Cell Line Encyclopedia (CCLE) dataset (*Barretina et al., 2012*). Here, *ITGA2* is the most prominent collagen-binding integrin expressed among the 47 reported EOC cell lines independent of their histological subtypes and *TP53* status (*Figure 2A*). The protein expression of collagen-binding integrins in eleven EOC cell lines coincided with the transcriptomic analysis in CCLE dataset (*Figure 2B*). Although high-grade serous ovarian cancer is the most common and lethal histotype accounting for 60–70% of EOC death (*Vaughan et al., 2011*), *ITGA2* mRNA expression of the EOC cell lines showed no significant correlation (R = −0.0012, p=0.99) to the suitability scores of serous histotype (*Domcke et al., 2013*; *Figure 2C*). This is in line with the EOC transcriptomic dataset (GSE51088) (*Karlan et al., 2014*; *Figure 2—figure supplement 1A*) and our tissue

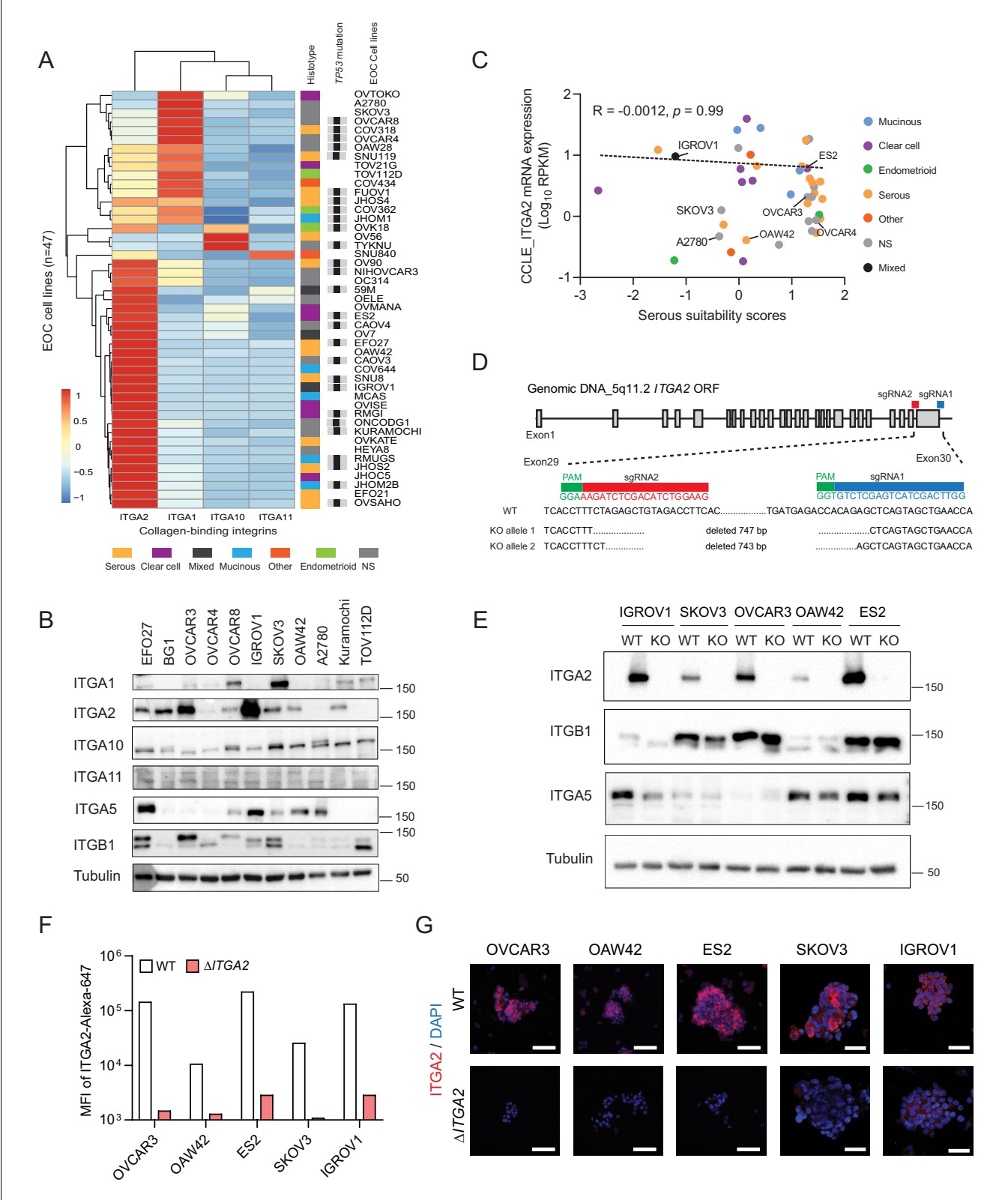

**Figure 2.** Distribution of collagen-binding integrin expression and generation of ΔITGA2 in EOC cell lines using CRISPR-Cas9. (A) Hierarchical cluster analysis and heat map visualization of known collagen-binding integrin-encoding genes among EOC cell lines using the Cancer Cell Line Encyclopedia (CCLE). The original histological subtypes and *TP53* genomic mutation status were annotated as previously described (*Domcke et al., 2013*). (B) Representative western blot showing expression of collagen-binding integrins among eleven EOC cell lines. (C) *ITGA2* expression (Spearman

*Figure 2 continued on next page*

*Figure 2 continued*

correlation R = −0.0012, p=0.99) is independent of histological subtypes and previously described serous suitability scores (*Domcke et al., 2013*). Representative EOC cell lines used in this study are annotated. (D) CRISPR-*Cas9* mediated deletion of *ITGA2* targeting exon 29–30, resulting in a 743 and 747 bp genomic deletion. (E) Immunoblot confirms loss of ITGA2 expression in five of Δ*ITGA2* (KO) cell lines while the binding partner ITGB1 was not altered. (F) Bar chart showed the median fluorescence intensity of ITGA2 expression in WT and Δ*ITGA2* cells by flow cytometry. (G) Representative fluorescence images with membranous ITGA2 expression in WT and Δ*ITGA2* cancer spheroids collected from ultra-low attachment plate. Scale bar 100 μm.

The online version of this article includes the following figure supplement(s) for figure 2:

**Figure supplement 1.** Heterogeneous ITGA2 expression in matched primary and metastatic tumor independent of tumor origin and histotypes.

**Figure supplement 2.** Heterogeneous *ITGA2* expression is independent of tumor histotypes and *TP53* status in ovarian cancer.

microarray data. We observed an overall high percentage of positive ITGA2 cytoplasmic (72.5%) and membranous staining (66%) of tumor cells (total n = 200 EOC patients) without differences for histological grade, FIGO stage, and subtypes (*Figure 2—figure supplement 1B–E*). However, an inconsistent pattern in this regards was observed for three additional transcriptomic datasets investigated (GSE73614; n = 107) (*Winterhoff et al., 2016*) GSE2109 (n = 204) and GSE9891 (n = 285) (*Tothill et al., 2008*; *Figure 2—figure supplement 2A*). We also confirmed that the mutational status of *TP53* does not correlate with ITGA2 expression in patient-derived samples (*Figure 2—figure supplement 2B–C*). Taken together, having investigated five independent transcriptomic data sets on ITGA2 gene and protein expression without revealing a consistent pattern throughout the majority of the studies suggested that collagen-ITGA2 interaction may act as universal phenomenon for peritoneal metastasis independent of EOC histotypes or genomic mutation of *TP53*. Therefore, we selected five representative EOC cell lines based on different histological subtypes as well as their endogenous ITGA2 expression levels (ITGA2^high to ITGA2^low: IGROV1, ES2, OVCAR3, SKOV3, and OAW42) for this study. We applied a paired-sgRNA-based CRISPR-*Cas9* strategy targeting the open reading frame of ITGA2 transmembrane domain (exon 29–30) at chromosome 5q11.2. The clonal selection of Δ*ITGA2* cells harboring a 743–747 bp homozygous deletion at the desired genomic locus was confirmed by genotyping PCR and Sanger DNA sequencing (*Figure 2D*). Loss of total ITGA2 expression in Δ*ITGA2* cells was verified by western blot (*Figure 2E*) and its associated receptor integrin β1 (ITGB1) was unaltered in all the Δ*ITGA2* cells. Moreover, loss of membranous ITGA2 expression in 2D cell culture was confirmed by flow cytometry (*Figure 2F*) and immunofluorescence for 3D culture of multicellular spheroids (*Figure 2G*).

## ITGA2 promotes cancer cell migration, anoikis resistance, and extravasation in vitro and in zebrafish embryo tumor xenograft

We next investigated the phenotypic changes between wildtype (WT) and Δ*ITGA2* cells to evaluate the biological functions of *ITGA2*. Of note, significantly reduced cell proliferation was observed in Δ*ITGA2* cells only on collagen-coated plates. In contrast, neither classical plastic nor fibronectin-coated plates showed such a difference (*Figure 3A–B* and *Figure 3—figure supplement 1*). Moreover, chemotaxis-induced cell motility was significantly reduced in Δ*ITGA2* cells (*Figure 3C*). Further investigation of cell detachment-induced anoikis, a critical step of tumor cells with enhanced mesenchymal features or undergoing malignant transformation, showed a significant increase of apoptotic (Annexin V^+) and dead (DAPI^+) cells in OVCAR3, SKOV3, and IGROV1 Δ*ITGA2* cells (*p<0.05; *Figure 3D*). The enhanced apoptosis in Δ*ITGA2* cells under non-adherent spheroid condition coincided with elevated levels of cleaved PARP, (*Figure 3E*) suggesting that ITGA2 promotes anoikis resistance in EOC cells. Similarly, we observed an increased expression of ITGA2 in spheroid as compared to adherent condition likely to respond the stress induced by ECM disengagement. Overall, the phenotypic changes of Δ*ITGA2* led us to study the anoikis resistance and early cell extravasation events in more detail. Thus, we transplanted CM-Dil-labeled WT and Δ*ITGA2* cells into transgenic Tg (*kdrl:eGFP*) zebrafish and analyzed survival and extravasated cancer cells 3 days post-injection as described previously (*Figure 3F*; *Jacob et al., 2018*). The percentage of in vivo survival of Δ*ITGA2* cells was significantly decreased compared to WT in all the tested EOC cells (*Figure 3G*), which is in line with the increased apoptosis of Δ*ITGA2* cells in the detachment-induced anoikis condition. Interestingly, we observed reduced tumor cell cluster formation in Δ*ITGA2* of IGROV1 and SKOV3 compared to WT cells (*Figure 3H-I*). The absence of tumor cell clusters and low number of extravasated

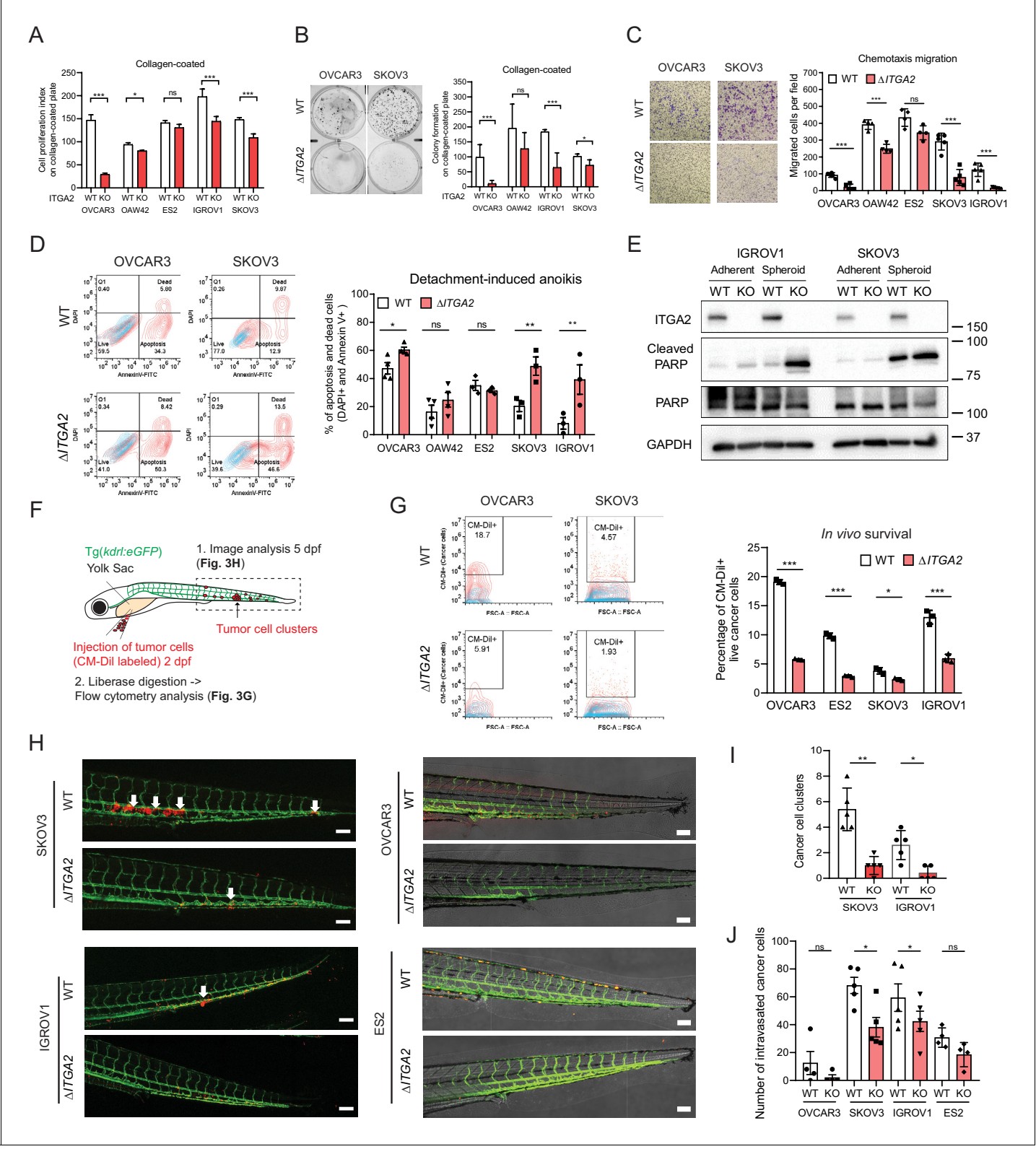

**Figure 3.** ITGA2 promotes EOC cell migration, anoikis resistance, and extravasation in vitro and in vivo. (**A**) Cell proliferation index of WT and Δ*ITGA2* cells in collagen-coated plate. (**B**) Anchorage-dependent colony formation assay for WT and Δ*ITGA2* cells on collagen-coated plate after 7 days incubation. Bar chart represents the mean ± SD of total colony counts. (**C**) Chemotaxis cell migration assay and quantification of cell migration (24 hr). Scale bar 200 μm. (**D**) Cell detachment-induced apoptosis (anoikis) assay. WT and Δ*ITGA2* cells stained with Annexin V-FITC and DAPI after 3–5 days of

*Figure 3 continued on next page*

Figure 3 continued

cultivation in ultra-low attachment plate to identify apoptotic dead cells (FITC$^+$/DAPI$^+$). Mean ± SD (*p<0.05 from two independent experiments. (**E**) Western blot shows increased cleaved PARP in the KO compared to WT cells under non-adherent spheroid condition. (**F**) Scheme of tumor cells transplant model in Tg(*kdrl*:eGFP) zebrafish. CM-Dil labeled cancer cells were transplanted into zebrafish at two dpf (days post fertilization). (**G**) At five dpf, fish were enzymatically dissociated to single cells and analyzed in vivo survival of cancer cells by flow cytometry. Representative counter plot shows the percentage of CM-Dil+ cancer cells. Bar chart summarizes data from n = 5 fish per group. (**H**) Representative confocal images of transplanted fish (n = 5 per group) showing the tumor cluster formation in proximity of the circulatory loop (white arrow) at five dpf. Scale bar 100 µm. (**I**) Numbers of cluster formation and (**J**) single cells in tail vein were counted and represented as mean ± SD (unpaired Student's t-test, *p<0.05).

The online version of this article includes the following figure supplement(s) for figure 3:

**Figure supplement 1.** The dependence of ECM in cell proliferation and colony formation assay.

cells in OVCAR3 xenograft might be explained by the generally low migratory phenotype of OVCAR3 and its distinct epithelial feature which is in concordance with the in vitro migration assay. Nevertheless, there is a similar trend of decreased extravasated single cells in the zebrafish Δ*ITGA2* xenograft, suggesting that ITGA2 promotes anoikis resistance and extravasation of EOC cells in vivo (*Figure 3J*).

## ITGA2 promotes cancer cell adhesion to collagen

In order to establish peritoneal metastasis, disseminated tumor cells require initial adhesion to the peritoneum and omentum, followed by invasion into the mesothelial monolayer and its underlying ECM network (*Kenny et al., 2014*). Here, we hypothesized that ITGA2$^+$ EOC cells disseminate and adhere to the omentum more efficiently than ITGA2$^-$ cells through cell-ECM interaction. Thus, we performed in vitro ECM adhesion assay by seeding suspended WT and Δ*ITGA2* cells on top of different ECM-coated (collagen type I, III, IV, laminin, and fibronectin) plate. Notably, the consistently strong cell adhesion efficiency to collagen type I, III, and IV in parental cell lines significantly diminished upon ITGA2 deletion. However, cell adhesion to laminin and fibronectin appeared in a cell line-dependent manner in Δ*ITGA2* cell lines which may be due to the redundancy of multiple integrin subunits. (*Figure 4A-B*). *Vice versa*, constitutive overexpression of ITGA2 in ITGA2$^{low}$ cancer cell line (n = 4) significantly enhanced cell adhesion to collagen I and IV (p<0.001; *Figure 4C* and *Figure 4— figure supplement 1A–C*) indicating a cell line-independent binding specificity of ITGA2 to collagen I, III, and IV. The increased spheroid formation in the ITGA2-OE cell lines and decreased spheroid formation in Δ*ITGA2* cells further supports our finding that ITGA2 promotes anchorage-independent growth (*Figure 4—figure supplement 1D–F*). Next, to characterize the dynamics of cell-collagen adhesion, we traced single cell movement, migration velocity, and focal adhesion by monitoring RFP-expressing WT and Δ*ITGA2* cancer cells on established bioprinted 3D-collagen slides. The single cell adhesion process was monitored by measuring the average surface area ratio (i.e. cells area at $t_n/t_0$) for up to 6 hr using time-lapse fluorescence confocal microscopy. We observed that Δ*ITGA2* cells failed to establish stable adhesion on collagen I as the surface area ratio remained <1 over time (*Figure 4D*). Likewise, ITGA2$^{low}$ A2780 cells were not able to adhere to collagen while ITGA2 overexpressing cells enhanced adhesion to collagen (*Figure 4D*). Cell-surface focal adhesion and expansion subcellular structure on bioprinted collagen was also visualized in WT cells but not in Δ*ITGA2* cells by scanning electron microscopy (SEM) analysis. (*Figure 4E*). The overall cell migration path on collagen was also significantly elevated in ITGA2$^+$ cells (*Figure 4F*), indicating that ITGA2 facilitates the dynamic interplay between cell-to-collagen adhesion and movement which may accelerate the passive dissemination in EOC metastasis.

## TC-I-15 treatment blocks patients-derived ATCs adhesion to collagens

To evaluate whether ITGA2$^{high}$ ATCs also show enhanced adherence capacity, we collected the ATCs from EOC patients (n = 20) and assessed the expression of ITGA2 and ITGA5 by flow cytometry (*Figure 5A*). The ATCs displayed a distinct expression pattern of ITGA2 and ITGA5, likely resulted in a different cell adhesion affinity to collagen or fibronectin. Notably, there is no significant correlation between ITGA2 and ITGA5 expression in CCLE dataset (*Figure 5—figure supplement 1*). The ECM ligand-receptor specificity was supported by a positive correlation between the percentage of ITGA2$^+$ ATCs and the adhesion efficiency to type I collagen (R = 0.6858, p=0.0288) but

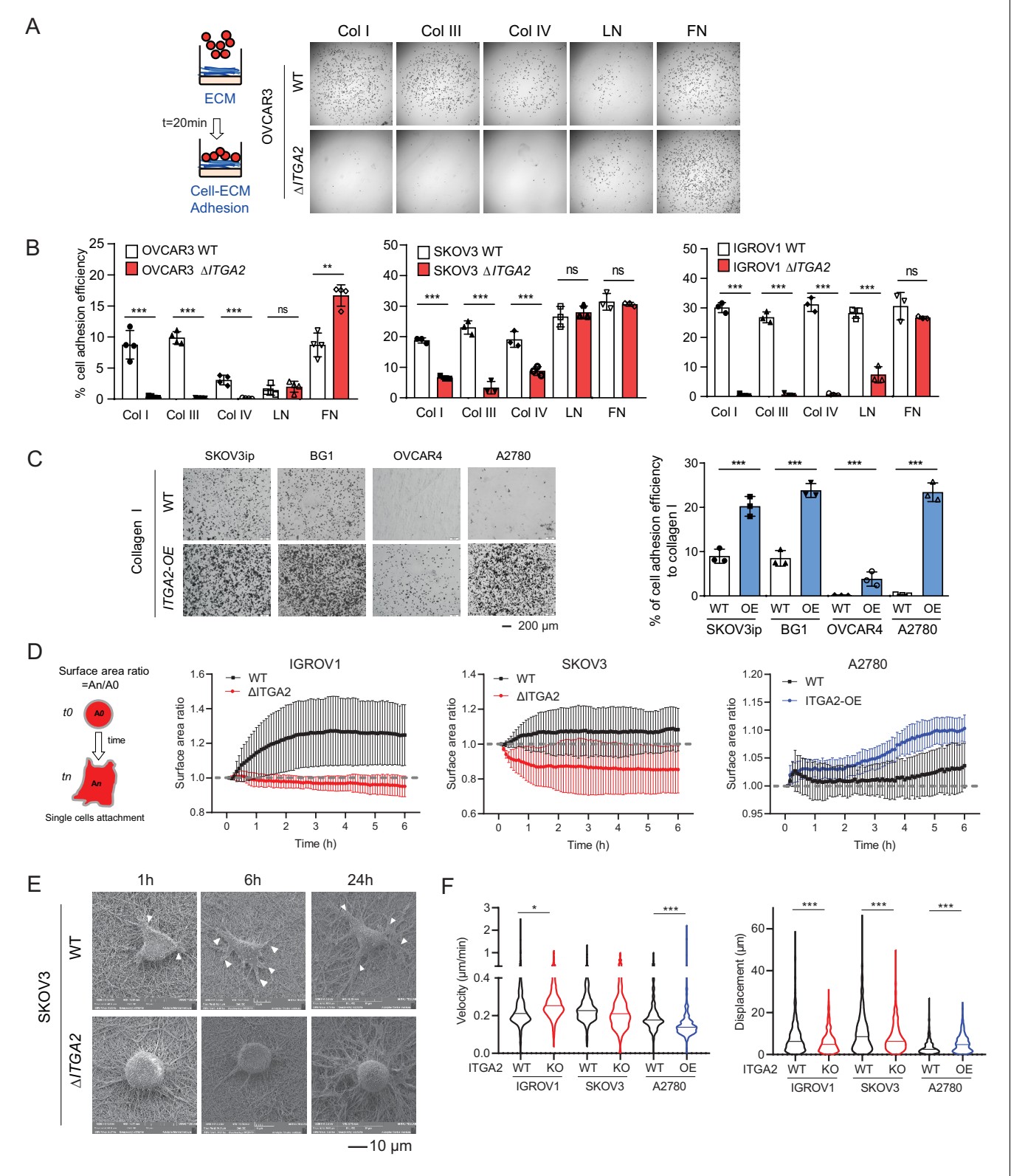

**Figure 4.** ITGA2 promotes cancer cell adhesion to collagen. (**A**) Static cancer cell-ECM adhesion assay. Representative images showing OVCAR3 WT and *ΔITGA2* cell adhesion to different ECM proteins, including collagen type I, III, IV, laminin, and fibronectin. (**B**) Bar charts with mean ± SD of the percentage of cell adhesion efficiency of OVCAR3, SKOV3, and IGORV1 cells from three independent experiments. (unpaired Student's t-test, ***p<0.001) (**C**) Representative images and bar charts showing the cell adhesion to collagen I in four ITGA2-overexpressed EOC cell lines (unpaired

*Figure 4 continued on next page*

*Figure 4 continued*

Student's t-test, ***p<0.001). Scale bar 200 µm. (D) Single cell adhesion on bioprinted collagen was monitored by time-lapse microscopy. Data represent the surface area ratio as mean ± SD (n > 300 single cells were analyzed per group). (E) Representative scanning electron microscope (SEM) images showing the SKOV3 WT and ΔITGA2 cellular adherence on bioprinted collagen at 1, 6, and 24 hr. White arrow indicates the polarized filopodia structure. Scale bar 10 µm. (F) Velocity and displacement of single cells on collagen-coated slides were measured (n > 300 per group from three independent experiments. Unpaired Student's t-test, *p<0.05, ***p<0.001).

The online version of this article includes the following figure supplement(s) for figure 4:

**Figure supplement 1.** Characterization of constitutive ITGA2 expression in selected ovarian cancer cell lines.

not to fibronectin (R = −0.04, p=0.902) (*Figure 5B*). *Vice versa*, our ΔITGA5 generated cell lines showed reduced cell to fibronectin adhesion with only marginal change for collagen binding (*Figure 5—figure supplement 2*). Furthermore, by treating EOC cell lines and patients-derived ATCs cells with the selective integrin α2β1 inhibitor TC-I-15. The inhibitor specifically blocked ITGA2 interaction to collagen with marginal difference to fibronectin (*Figure 5C-E*). This inhibitory effect was not due to cytotoxicity of the drug itself since a long-term treatment (10 days) with the TC-I-15 inhibitor showed no significant impact on cell proliferation and spheroid formation (*Figure 5F*). Taken together, our findings suggest that ITGA2 is essential in mediating the initial cancer cell-collagen adhesion, a key step for peritoneal dissemination. TC-I-15 specifically blocks ITGA2-collagen interaction may therefore impede cancer cells dissemination to the collagen-rich omentum.

## ITGA2 promotes mesothelial clearance and peritoneal metastasis in vivo

After an initial tumor cell-collagen adhesion, a subsequent cell–cell adhesion and invasion through the mesothelial monolayer is a prerequisite to establish tumor implants in the omentum or peritoneum. Earlier studies suggest that ovarian cancer spheroids use fibronectin-binding integrin α5- and talin-dependent traction force to displace mesothelial cells and gain access to the underlying ECM (*Davidowitz et al., 2014*; *Iwanicki et al., 2011*). To identify whether ITGA2 facilitates cancer cell adhesion to mesothelial cells, we performed a co-culture in vitro competition adhesion assay. Equal numbers of fluorescence-labelled WT and ΔITGA2 cells were seeded onto a confluent mesothelial monolayer, adhesion was allowed for different times and then harvested for subsequent flow cytometry analysis. The ratio of cancer cell adhesion to mesothelial cells was significantly increased for 12 and 24 hr incubation (1.92- to 2.59-fold; WT/ ΔITGA2 cells) (*Figure 6A*), supporting the role of ITGA2 in facilitating mesothelial cell adhesion. In addition to single cell-mediated adhesion, we also evaluated the invasiveness of cancer spheroids to mesothelial cells following the dynamics of mesothelial clearance. Constitutively EGFP-expressing mesothelial cells (MeT-5A) were cultured on different ECM components and allowed to form a confluent monolayer before co-culture with RFP-expressing ovarian cancer spheroids. Among the different types of ECM components, collagen markedly enhanced cancer spheroid invasion through the mesothelial monolayer (*Figure 6B*) which further supports the premetastatic role of collagen in cancer cell dissemination. Furthermore, we evaluated cancer spheroid-mediated mesothelial clearance activity of different cell lines including WT, ΔITGA2 (n = 4), and ITGA2-OE cells (n = 4). The mesothelial clearance ratio of ΔITGA2 spheroids was significantly reduced compared to WT cells in IGROV1, OAW42, and ES2 cells (*Figure 6C-D*) and *vice versa* markedly increased in ITGA2-OE spheroids in SKOV3ip cells (*Figure 6E*). However, ITGA2-OE marginally promoted mesothelial clearance in BG1, OVCAR4, and A2780 cells. Such observation might be explained by ITGA2-independent mechanism involved in mediated mesothelial clearance.

To determine whether ITGA2 promotes cancer cells to selectively metastasize to the peritoneum and omentum, SKOV3 WT and ΔITGA2 cells were injected intraperitoneally into NIH(S)II: nu/nu mice. Histological examination of peritoneal organs after 8 weeks revealed a significant reduction of total tumor implants in ΔITGA2 xenografts compared to WT xenografts (*Figure 6F*). Moreover, a systematic microscopic examination revealed that the omentum harbors the highest number of metastatic foci compared to other organs in the peritoneal cavity reflecting the clinical observation that

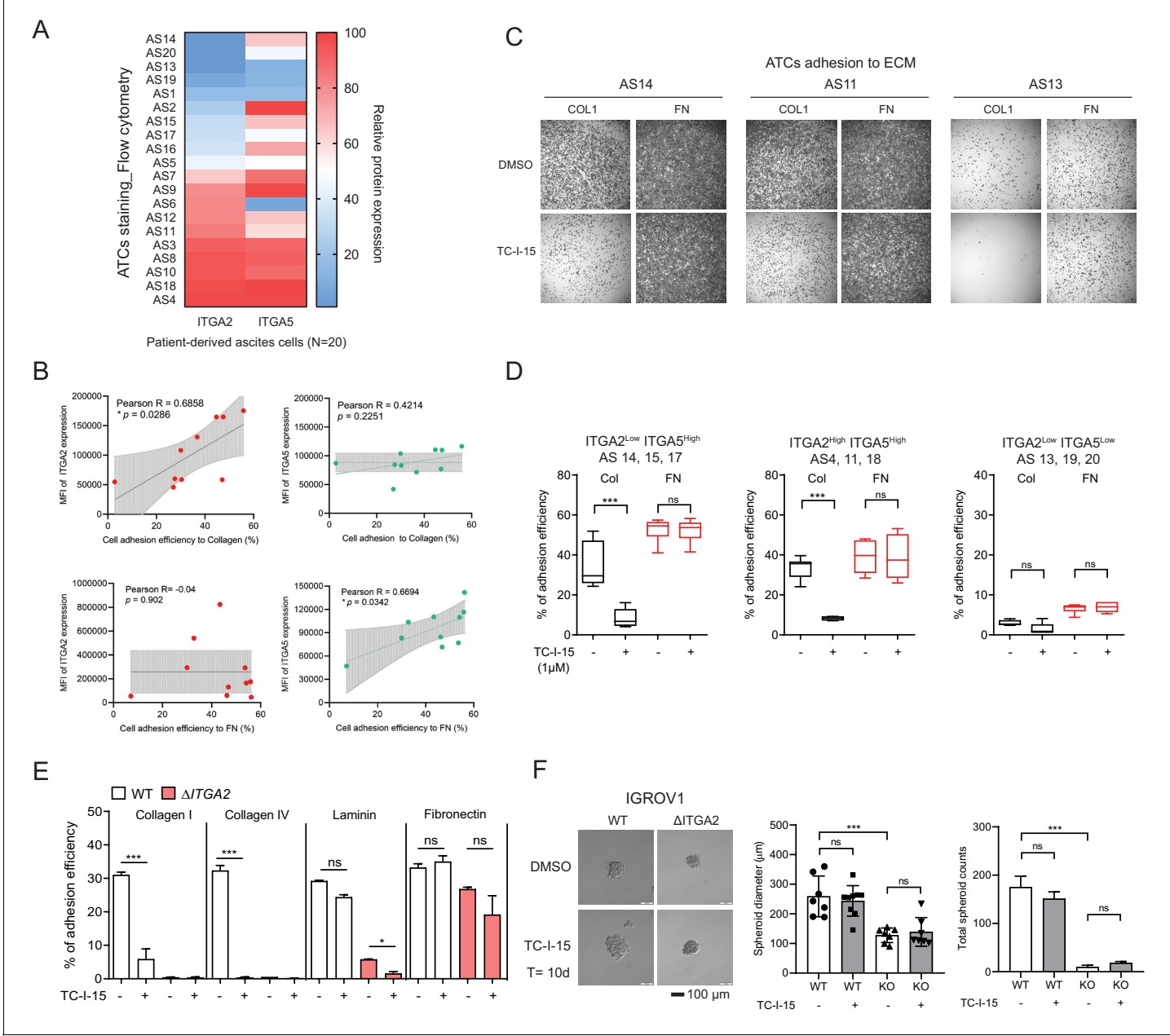

**Figure 5.** ITGA2 inhibitor selectively blocks ATCs and cancer cell adhesion to collagen. (**A**) Flow cytometry analysis of ITGA2 and ITGA5 expression from patient-derived ATCs. (**B**) Pearson's correlation of ITGA2 or ITGA5 expression and cell-to-collagen/fibronectin adhesion efficiency (n = 10). (**C**) ATC adhesion to collagen and fibronectin was performed after 20 mins pretreatment of 1 μM TC-I-15 or DMSO as control. Representative images show that TC-I-15 inhibits primary ATCs adhesion to collagen I but not fibronectin. (**D**) Box-whisker plots show the percentage of ATCs adhesion efficiency to collagen and fibronectin, respectively. (unpaired Student's t-test, ***p<0.001). (**E**) Percentage of IGROV1 cell-ECM adhesion with (+) or without (-) 20 mins pretreatment of TC-I-15. (**F**) Anchorage-independent cell growth in the presence or absence of TC-I-15 inhibitor for 10 days (TC-I-15 containing medium was refreshed every 2 days). Bar chart shows the mean ± SD of spheroid diameters and spheroid number counts (***p<0.001) in IGROV1 WT and ΔITGA2 cells. Scale bar 100 μm.

The online version of this article includes the following figure supplement(s) for figure 5:

**Figure supplement 1.** *ITGA2* expression does not correlate with the expression of *ITGA5*.
**Figure supplement 2.** Establishment and characterization of *ITGA5* knockout ovarian cancer cell lines.

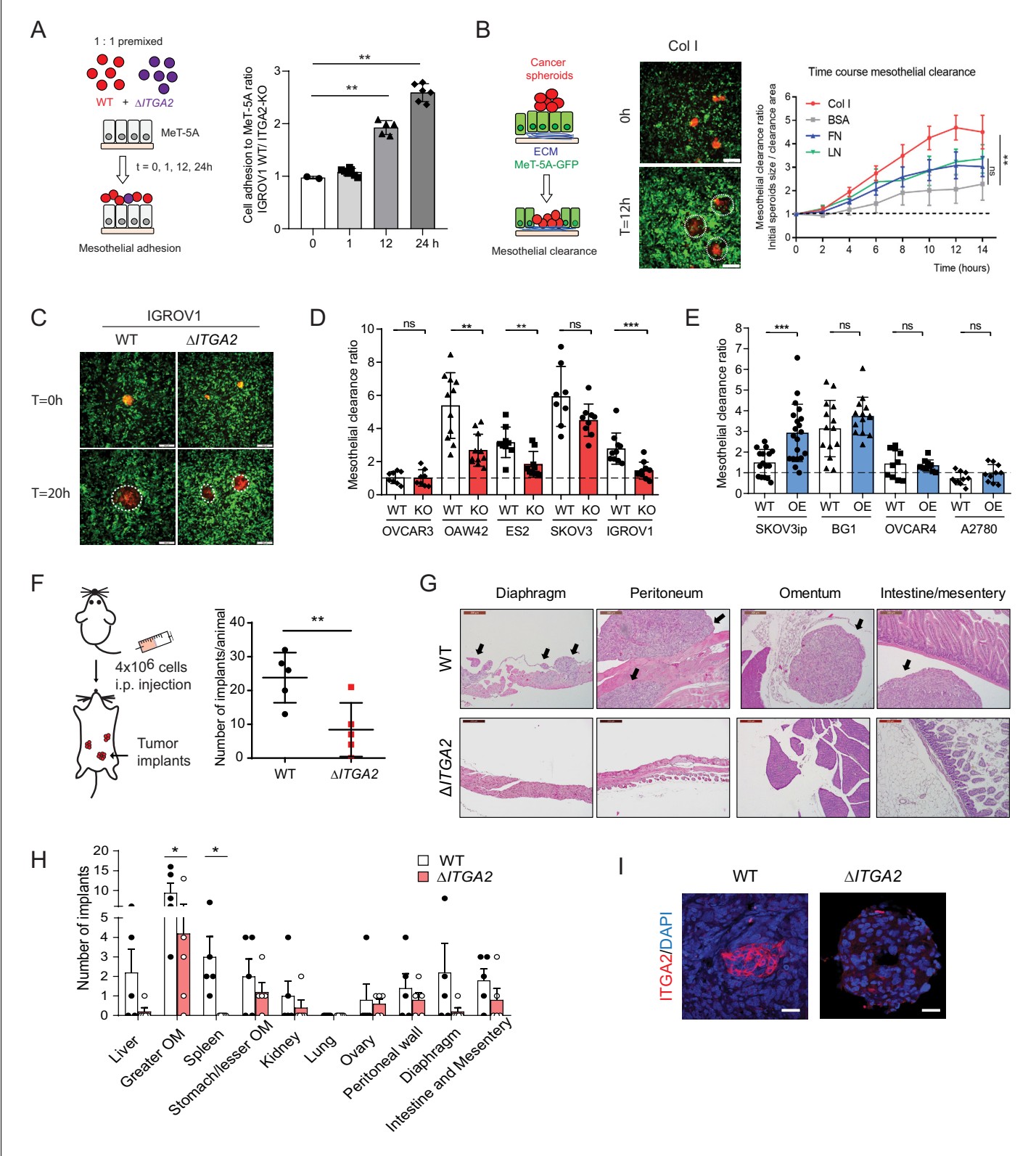

**Figure 6.** ITGA2 promotes mesothelial clearance and metastasis to the omentum in vivo. (**A**) Cancer cell-mesothelial adhesion assay was performed by seeding equal numbers of fluorophore-labeled WT and *ΔITGA2* single cancer cells on a confluent monolayer of MeT-5A cells. Bar chart shows the adhesion ratio of WT vs *ΔITGA2* to MeT-5A cells at different time points. (**B**) Stable RFP-expressing cancer cell spheroids were seeded on top of a GFP-expressing MeT-5A mesothelial monolayer pre-coated with different ECM substrates. Mesothelial clearance activity was measured over time by time-

*Figure 6 continued on next page*

*Figure 6 continued*

lapse microscopy. Clearance ratio >1 suggests active enhanced clearance activity. Scale bar 200 µm. Line chart summarizes mesothelial clearance activity on different ECM substrates over time. (C) Representative images of mesothelial clearance assay of SKOV3 WT and ΔITGA2 cancer cell line at time point 0 and 20 hr. Scale bar 200 µm. (D) Bar chart shows mesothelial clearance ratio for WT *versus* ΔITGA2 (n = 5) and (E) WT *versus* ITGA2-overexpressed (OE) (n = 4) EOC cell lines with mean ± SD (unpaired Student's t-test, ***p<0.001), each dot represents single spheroid clearance activity. (F) Dot plot shows the total number of tumor implants per animal after 8 weeks intraperitoneal injection of $4 \times 10^6$ SKOV3 WT and ΔITGA2 cells in NIH(S)II: nu/nu mice. (G) Representative H and E staining of the xenografts and metastases. Black arrows indicate the tumor metastases. Scale bar 200 µm. (H) Bar chart summarized the number of tumor foci in different organs. Mean ± SD (One-way Anova, *p<0.05). (I) Representative immunofluorescence staining of ITGA2 expression in the omental tumor xenograft.

omentum is the most frequent site of metastasis (*Figure 6G and H*). The significant reduction of metastatic foci in the omentum of ΔITGA2 xenografts further supported the role of ITGA2 in promoting peritoneal dissemination and invasion into collagen-rich omentum (*Figure 6H and I*). Collectively, our data revealed that ITGA2 may contribute to the early onset of peritoneal metastasis by triggering single cells and spheroids adhesion to collagen, mesothelial layer and subsequent mesothelial invasion.

## Identification of ITGA2-dependent signaling pathway using phosphoproteomics

In order to understand the molecular mechanism underlying ITGA2-dependent signaling, we analyzed the total proteome and phosphoproteome of ΔITGA2 and WT EOC cell lines (n = 3) (*Figure 7—source data 1*) in the presence of collagen. We identified 69 downregulated phosphoproteins shared across all three ΔITGA2 cell lines (*Figure 7A*, *Supplementary file 1b*). Next, by applying the gene ontological (GO) analysis, we classified that the enriched cell components in the biological process were related to cell organization, polarity, and GTPase activity (*Figure 7B*). Such observation was further supported by Gene Set Enrichment Analysis (GSEA) and Reactome pathway analysis showing that MAPK1/3 activation and cell junction organization are significantly regulated through the ITGA2 axis (*Figure 7C–D*). Moreover, IGROV1 ΔITGA2 and WT cell was further analyzed because of the highest phosphoproteins detection in the mass spectrometry with a total of 18,471 phosphopeptides belonging to 3928 phosphoproteins. A total of 1061 phosphoproteins were significantly downregulated in ΔITGA2 (*Figure 7E*, and *Supplementary file 1c*). Unsupervised hierarchal clustering of the significantly altered phosphoproteins showed 20 downregulated proteins involved in the KEGG focal adhesion pathway (hsa04510) (*Figure 7F*). The KEGG pathview package (*Luo and Brouwer, 2013*) allowed us to visualize the differential expression of phosphoproteins in the focal adhesion signaling (e.g. downregulation of Src-FAK-sos-Raf-ERK1/2 cascades in ΔITGA2 cells) (*Figure 7—figure supplement 1A*). In addition, transcriptomic analysis of the TCGA-ovarian cancer dataset revealed a positive correlation of *ITGA2* expression to *COL1A2*, *COL3A1*, and *COL5A1* involved in focal adhesion pathway (*Figure 7—figure supplement 1B*). In line with a previous finding showing that ITGA2 positively regulates the transcription of fibrillary collagens (*Ivaska et al., 1999*), the ITGA2-collagen axis would likely constitute a positive feed-back loop between tumor cells and stromal cells in the metastatic niche.

On the other hand, a significant enrichment of cell junction organization and cell–cell communication pathways was upregulated in ΔITGA2 cells (*Figure 7D*) suggesting a downregulation of epithelial-mesenchymal transition (EMT) upon genetic disruption of ITGA2. Consistent with proteomic profiling and western blot analysis, we confirmed that cell junction adhesion proteins including E-cadherin and EpCAM were significantly upregulated in the ΔITGA2 EOC cells (*Figure 7—figure supplement 1C*). Such enhanced expression of epithelial-associated proteins and cell–cell interaction may also explain the reduction of cell migration and invasion properties of the ΔITGA2 cells which is also support by recent studies showing that silencing ITGA2 restored E-cadherin expression and downregulated mesenchymal marker in chemo-resistant gastric cancer cells (*Wang et al., 2020*) as well as exosome-mediated transfer of ITGA2 in prostate cancer cells (*Gaballa et al., 2020*).

Multiple differential phosphorylation sites with relevant functional biological consequences were also identified in our analysis. In line with the phenotypic change of ΔITGA2 cells, significantly

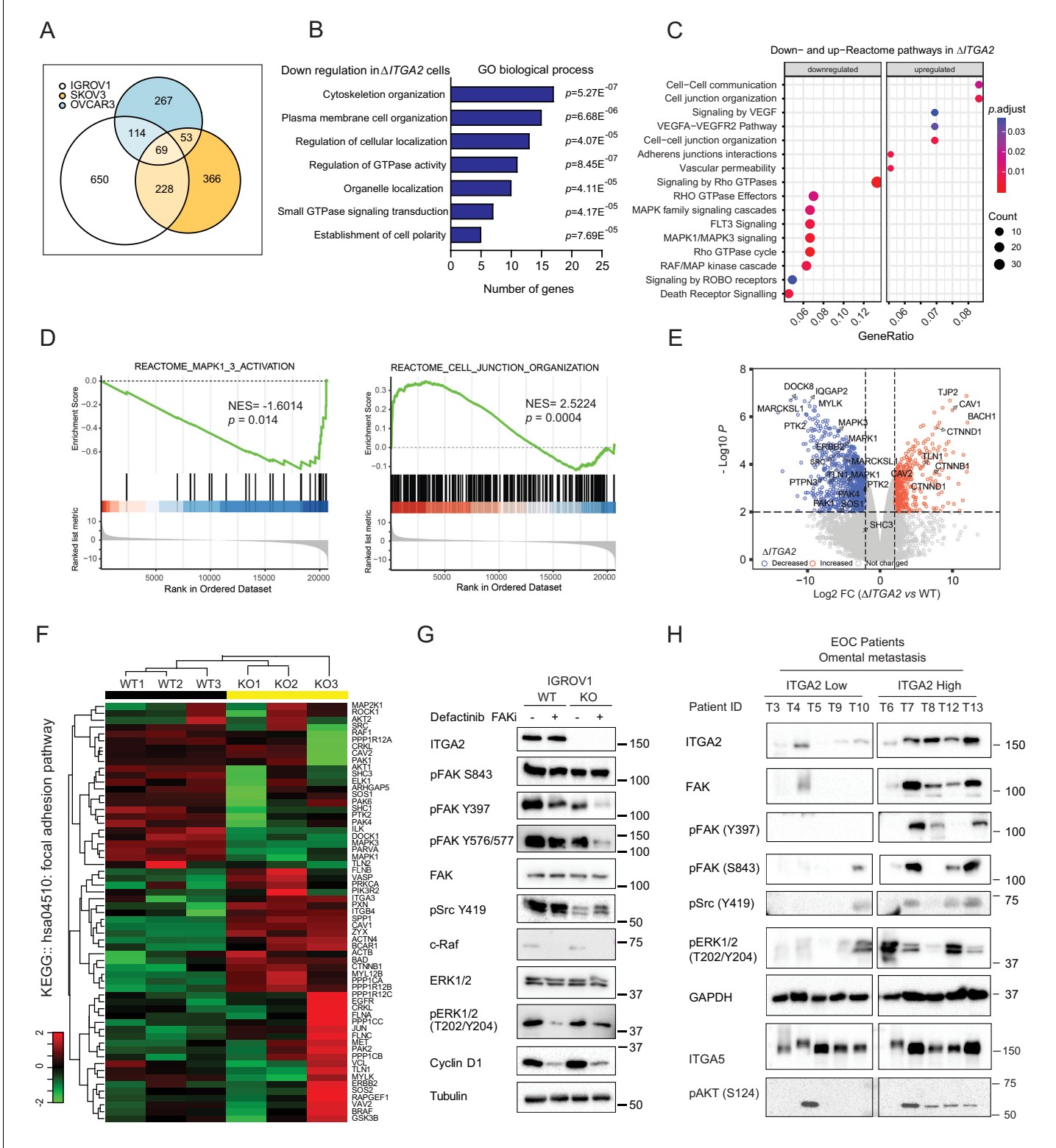

**Figure 7.** Identification of ITGA2-dependent activation of FAK and MAPK signaling axis using phosphoproteomics. (**A**) Venn-diagram highlights the number of significantly downregulated phosphoproteins shared among three Δ*ITGA2* ovarian cancer cell lines. (**B**) Top seven gene ontology (GO) biological process terms with highest statistical significance. The horizontal axis displays the number of genes in the intersection group of three Δ*ITGA2* ovarian cancer cell lines. (**C**) GSEA analysis identified up- or downregulated pathways in Δ*ITGA2* cells. The Cluster Profiler dot plot visualization shows enriched terms as dots. The highest ranking of the significantly enriched pathway is displayed. (**D**) Reactome pathway analysis of MAPK and cell

*Figure 7 continued on next page*

*Figure 7 continued*

junction organization pathway comparing Δ*ITGA2* with WT cells. NES = normalized enrichment score. Nominal p-value calculated by the permutation test in GSEA analysis. (**E**) Volcano plot shows differentially expressed phosphoproteins between WT and Δ*ITGA2* cells. y-axis defines the statistical significance -Log10p<2 and x-axis defines the magnitude of Log2-fold change >2 or < −2. Colored circle defines significantly decreased (blue) or increased (red) phosphopeptides in Δ*ITGA2* cells. (**F**) Heatmap analysis of KEGG hsa04510:focal adhesion pathway phosphoproteins. Data were analyzed based on three biological replicates of IGROV1 WT and Δ*ITGA2* cells. (**G**) Western blot analysis shows major change of phosphoproteins in the focal adhesion signaling upon treatment of 5 µM Defactinib for 24 hr. (**H**) Western blot analysis shows induced activation and phosphorylation of FAK/Src/MAPK signaling in patient-derived ITGA2$^{high}$ compared to ITGA2$^{low}$ omental tumor (n = 10).

The online version of this article includes the following source data and figure supplement(s) for figure 7:

**Source data 1.** Analysis of phosphoproteomics in IGROV1, SKOV3, and OVCAR3 WT *versus* Δ*ITGA2* cell lines.
**Figure supplement 1.** Phosphoproteomics identify ITGA2-dependent activation of the FAK and MAPK signaling axis.
**Figure supplement 2.** Functional inhibition targeting ITGA2-dependent FAK/MAPK signaling axis.
**Figure supplement 3.** Functional inhibition of ITGA2-dependent FAK/MAPK signaling axis reduces spheroid formation, mesothelial clearance activity but not cell adhesion.

decreased phosphorylation of MAPK3$^{Y204}$ and MAPK1$^{T185/Y187}$, previously showing to inhibit its nuclear translocation and transactivation of cell proliferation signaling were identified in Δ*ITGA2* condition (*Figure 7G*, *Figure 7—figure supplements 1D* and *2*). The phosphorylation of AKT$^{S124}$ and AKT$^{S129}$ are also significantly reduced in the Δ*ITGA2* cells (*Figure 7—figure supplement 1D*). Both phosphorylation sites are reported in enhancing catalytic activity of AKT and promotes cell survival, suggesting a crosstalk between AKT and ITGA2-dependent signaling (*Di Maira et al., 2005*). The phosphorylation of the proto-oncogene tyrosine kinase p-Src$^{Y419}$ as well as FAK at site Y397, Y577, S722, and S843 were also significantly downregulated in Δ*ITGA2* condition (*Figure 7G*). The oncogenic role for FAK has been shown to sustain tumor spheroid proliferation and chemo-resistance in ovarian cancer (*Diaz Osterman et al., 2019*). FAK$^{Y397}$ autophosphorylation is a hallmark of FAK activity (*Kleinschmidt and Schlaepfer, 2017*) and collaborates with Src-dependent phosphorylation at FAK$^{Y576/Y577}$ leads to enhance adhesion-induced FAK activation. In order to evaluate whether inhibiting FAK phosphorylation would counteract with ITGA2-dependent spheroid formation and anoikis resistance, we tested FAK inhibitor in various in vitro assays. The FAK specific inhibitor Defactinib, previously showing initial signs of activity in relapsed EOC patients in combination with paclitaxel in a clinical phase I/Ib trial (*Patel et al., 2014*) was tested in this study. Here, Defactinib treatment in EOC cells showed no impact on inhibiting the upstream p-Src$^{419}$ but resulted in a significant reduction of p-FAK$^{Y397}$, p-FAK$^{Y576/577}$, and the downstream c-Raf, G1/S phase transition cyclin D1 expression (*Figure 7G*, *Figure 7—figure supplement 2*). The inhibition of FAK phosphorylation by Defactinib led to a dramatic decrease in cell survival, anoikis resistance, and spheroid formation (*Figure 7—figure supplement 3A–D*). Blockade of either FAK or MAPK signaling axis showed no direct impact of cell-collagen adhesion but significantly reduced mesothelial clearance activity of EOC cells, supporting the functional involvement of ITGA2, and FAK/MAPK axis in ovarian cancer metastasis model (*Figure 7—figure supplement 3E–F*). It is reasonable that the FAK/MAPK-dependent cell survival is downstream of integrin receptor signaling, therefore ITGA2-mediated initial cell adhesion is marginally affected. In line with phosphoproteomics and in vitro inhibition models, the EOC patients' samples also revealed a remarkable upregulation of phosphorylated FAK$^{Y397,S843}$, and ERK1/2 $^{T202/Y204}$ in ITGA2$^{High}$ omental metastases (*Figure 7H*). Taken together, our data suggest that the ITGA2-collagen axis triggers activation of FAK and MAPK signaling and supports cell survival, anoikis resistance, cell adhesion.

## Discussion

Given the significance of collagens comprising the major component of the ECM in the omentum, we investigated the role of collagen-binding integrins in the early onset of omental metastasis. There have been four collagen-binding integrin complexes (α1β1, α2β1, α10β1, and α11β1) implicated in a variety of malignancies with promoting activity in cancer cell dissemination (*Zeltz and Gullberg, 2016*). The transcriptomic analysis from CCLE database suggested that ITGA1 and ITGA2 are the

major collagen-binding integrins in EOC cell lines (*Figure 2A*). Analysis of protein expression shows only ITGA2 is significantly elevated in omental metastases compared to the normal omentum, suggesting its potential role in mediating metastatic events. High expression of ITGA2 has been reported to exacerbate experimental metastasis in melanoma, gastric, and colon cancer (*Baronas-Lowell et al., 2004*; *Bartolomé et al., 2014*; *Matsuoka et al., 2000*). Additional studies revealed a ligand-mediated dynamic function of ITGA2 in promoting hepatic metastasis through binding to collagen IV in liver (*Yoshimura et al., 2009*) as well as driving collagen-rich bone metastasis in a prostate tumor xenograft model (*Hall et al., 2006*). However, Ramirez et al. demonstrated that ITGA2 acts as a metastasis suppressor by increasing tumor cell intravasation in *Itga2* deficient mice (*Ramirez et al., 2011*).

In regards to ovarian cancer, a recent study demonstrated that ovarian cancer cells with elevated ITGA2 led to poor prognosis and survival. Overexpression of ITGA2 promoted in vitro cancer cell aggressiveness and in vivo tumor growth through regulation of the phosphorylation of forkhead box O1 (FoxO1) and AKT phosphorylation, which provided the mechanism for ITGA2-mediated paclitaxel-resistance in ovarian cancer (*Ma et al., 2020*). However, it remains uncertain how ITGA2 regulates AKT/Foxo1 axis and how ITGA2 is regulated in the ovarian cancer cells. Another study by *Shield et al., 2007* showed that enhanced cellular expression of integrin α2β1 influence spheroid disaggregation and proteolysis responsible for the peritoneal metastases. Functional blockade with monoclonal antibodies binding to α2, β1, and α2β1 integrin inhibited disaggregation and activation of MMP2/MMP9 in cancer spheroids. Although the participation of specific integrins in spheroid formation is not fully characterized, it is reasonable that in a spheroid scenario is more related to cell–cell rather than cell-ECM interaction that will influence integrin expression profile. Taken together, a comprehensive understanding of integrin-ECM interaction, a crucial step of cancer cell dissemination, is uncertain but of immense importance in order to prevent peritoneal metastasis.

In this study, we report a high abundance of ITGA2 in more than 60% of patients investigated present in primary, ATCs, and omental metastases of the investigated cancer samples, suggesting a selective potential of tumor cell adherence to the collagen-rich omentum. We propose that ITGA2 facilitates early events of peritoneal metastasis by enhancing anchorage-independent cell growth, cell adhesion to collagen, and displacement of mesothelial layer leading to peritoneal metastasis both in vivo and in vitro and by using several *ITGA2* knockout (n = 5) and overexpressed (n = 4) cell lines representing various histological histotype of EOC. Additionally, upregulation of multiple ECM proteins including COL1A1, COL1A2, COL3A1, COL5A1, COL11A1, FN1, and VCAN was also observed in omental metastases. ECM proteins have been associated with cancer progression and poor prognosis as shown by multilevel evidences from transcriptomics, proteomics, digital histopathology data (*Pearce et al., 2018*), as well as in a pooled hazard ratio model (*Figure 1—figure supplement 1D*). For example, COL1A1 promotes primary ovarian cancer cell adhesion and initiates tumor growth (*Burleson et al., 2004*; *Moser et al., 1996*). Hypoxia stimulates collagen remodeling by mesothelial cells and can be pharmacologically inhibited to reduce tumor burden in murine omental metastases (*Natarajan et al., 2019*). Furthermore, the enhanced expression of COL5A1, COL11A1, and VCAN through regulation of TGF-β1 signaling was identified in ovarian cancer metastasis. Consistently, elevated COL11A1 expression is also a predictor for poor survival in patients with serous ovarian cancer (*Cheon et al., 2014*; *Jia et al., 2016*).

In the present study, the origin of high COL1A1, COL1A2, COL3A1 expression identified in the normal omentum and omental metastases positively correlates with fibroblast activation protein-1α (FAP) and α-smooth muscle actin (ACTA2) (*Figure 1—figure supplement 1C*). Considering that fibroblasts possess very high contractile ability, promote angiogenesis, and stimulate epithelial tumor growth via secretion of growth factors and a variety of ECM, this might explain the correlation with poor prognosis of patients with elevated cancer-associated fibroblasts and collagens (*Gao et al., 2019*; *Pearce et al., 2018*). Intriguingly, the matrix metalloproteinases (e.g. MMP2, 9, and 14) capable of degrading collagens and proteoglycans were also found to be upregulated in omental metastases (*Figure 1A*, and *Supplementary file 1a*), consistent with the idea that ECM degradation and remodeling are essential steps for cancer cell invasion (*Shield et al., 2007*). On the other hand, downregulation of the matrix proteins in omental metastases including LAMA4, LAMB1, LAMC1, TNXB, and COL15A1 (*Figure 1A*) is associated with the integrity of mesothelial basement membrane and tissue homeostasis (*Pearce et al., 2018*). Peritoneal metastasis is common in patients with adenocarcinomas of ovarian, tubal, and peritoneal origin and underlying biological mechanisms

have been mainly focusing on an integrin α5β1-fibronectin interaction. Kenny et al. has demonstrated that the interaction of integrin α5β1-fibronectin between cancer cells and mesothelial cells can be blocked by a monoclonal antibody, resulting in lesser peritoneal metastasis (*Kenny et al., 2014*). Furthermore, a recent study highlighted the role of fibroblast-secreting EGF in promoting ITGA5high spheroid-mediated peritoneal dissemination (*Gao et al., 2019*). However, a recent phase II trial using the anti-α5β1 integrin antibody, Volociximab, showed no clinical improvement in patients with advanced ovarian cancer, and 75% of the patients experienced study-related adverse events (*Bell-McGuinn et al., 2011*; *Ricart et al., 2008*). Here, the redundancy of overlapping ligand specificities between integrins may be a possible escape mechanism of tumor cells. Therefore, our finding provides an alternative therapeutic potential of blocking ITGA2-mediated collagen adhesion using a selective α2β1 integrin antagonist, TC-I-15. Considering that cancer cell dissemination is the initial event of peritoneal metastasis, it is reasoned that the proposed integrin blockade may not be effective when metastasis has already taken place. As a result, a maintenance therapy after cytoreductive surgery might be more favorable to delay tumor recurrence.

Mechanistically, we have identified that ITGA2-collagen interaction fosters cancer cell survival, cell migration, and anoikis resistance through the FAK/MAPK signaling axis. As a result, targeting integrin-dependent downstream effectors may also provide a similar inhibitory effect as integrin inhibition to prevent cancer cell dissemination. Increased expression of FAK signaling proteins has been identified in omental metastases, which is linked to poor prognosis in EOC (*Nolasco-Quiroga et al., 2019*; *Sood et al., 2004*). In agreement with previous findings that depletion of FAK also inhibits local invasion and metastasis in vivo (*Provenzano et al., 2008*), we could demonstrate that FAK inhibitor Defactinib dramatically reduces cancer cell proliferation, spheroid formation, and mesothelial clearance. Moreover, the growth factor family receptors (e.g. EGFR, VEGFR) can also induce the FAK/Src/MAPK activation in an integrin-independent manner (*Bromann et al., 2004*), suggesting that targeting FAK may provide an ancillary effect to inhibit the growth of tumor cells.

Overall, we describe the critical contribution of collagens elevated in the premetastatic niche and the mechanism of how ITGA2-mediated cancer cell dissemination during peritoneal metastasis. The ITGA2-dependent phosphorylation of FAK and ERK1/2 triggers focal adhesion and MAPK signaling, leading to increased anoikis resistance and mesothelial invasion of cancer cells (*Figure 8*). The herein provided in vitro, in vivo, and ex vivo evidence suggests that selectively blocking cancer cell-to-collagen interaction or targeting FAK/MAPK-mediated signaling may present another opportunity for therapeutic intervention of peritoneal metastasis.

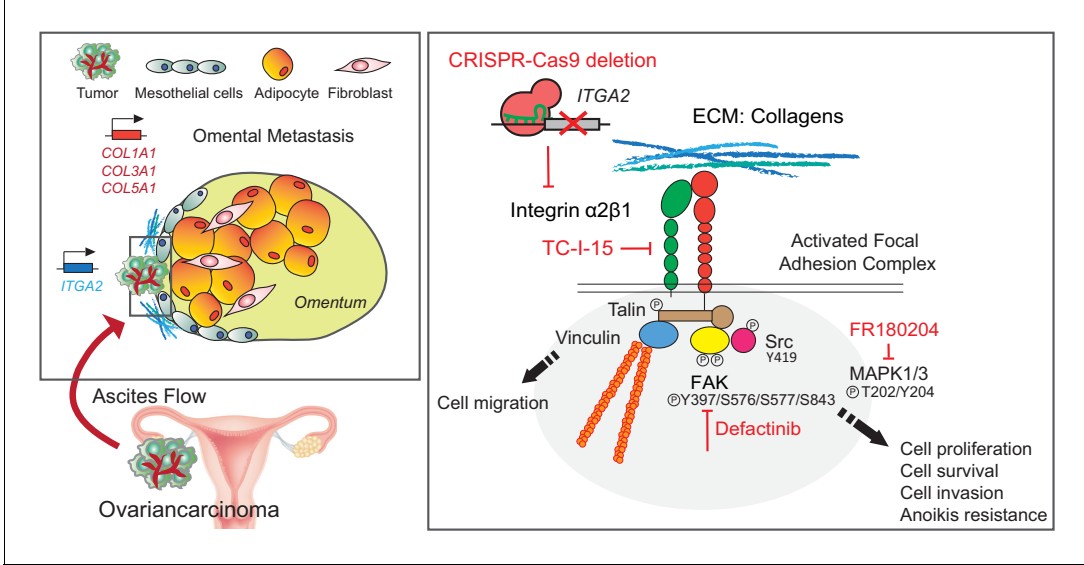

**Figure 8.** Schematic representation of the ITGA2-collagen dependent signaling axis in ovarian cancer metastasis.

# Materials and methods

## Key resources table

| Reagent type (species) or resource | Designation | Source or reference | Identifiers | Additional information |
|---|---|---|---|---|
| antibody | anti-human COL1A1 antibody (rabbit polyclonal) | Sigma-Aldrich | Cat# HPA011795, RRID:AB_1847088 | IHC (1:100) |
| antibody | Integrin β1 (rabbit monoclonal) | Cell Signaling Technology | Cat# 9699, RRID:AB_11178800 | WB (1:1000) |
| antibody | Integrin β1 antibody [12G10] (mouse monoclonal) | Abcam | Cat# ab30394, RRID:AB_775726 | WB (1:1000) |
| antibody | Integrin β1 (CD29)-BV510 (mouse monoclonal) | BD Biosciences | Cat# 747747 RRID:AB_2868388 | FACS (1:100) |
| antibody | Alexa Fluor 647 anti-human CD49a (mouse monoclonal) | BioLegend | Cat# 328310, RRID:AB_2129242 | FACS (1:100) |
| antibody | Anti-human Integrin α1/CD49a (mouse monoclonal) | R & D Systems | Cat# MAB5676, RRID:AB_10719143 | WB (1:500) |
| antibody | Anti-human Integrin α2 (rabbit monoclonal) | Abcam | Cat# ab181548, RRID:AB_2847852 | WB (1:5000) |
| antibody | FITC anti-human CD49b, Integrin α2 (mouse monoclonal) | BD Biosciences | Cat# 555498, RRID:AB_395888 | FACS (1:100) |
| antibody | Alexa Fluor 647 anti-human CD49b, Integrin α2 (mouse monoclonal) | BD Biosciences | Cat# 564314, RRID:AB_2738739 | FACS (1:100) |
| antibody | anti-human Integrin α5 [C9] (mouse monoclonal) | Santa Cruz Biotechnology | Cat# sc-376199, RRID:AB_10987904 | WB (1:200) |
| antibody | Alexa Fluor 647 anti-human CD49e Integrin α5 (mouse monoclonal) | BD Biosciences | Cat# 563578, RRID:AB_2738289 | FACS (1:100) |
| antibody | anti-integrin α10 (rabbit polyclonal) | Millipore | Cat# AB6030, RRID:AB_10806096 | WB (1:1000) |
| antibody | anti-human Integrin α11 (goat polyclonal) | R & D Systems | Cat# AF4235, RRID:AB_10644168 | WB (1:1000) |

*Continued on next page*

*Continued*

| Reagent type (species) or resource | Designation | Source or reference | Identifiers | Additional information |
|---|---|---|---|---|
| antibody | p44/42 MAPK (Erk1/2) Antibody (rabbit polyclonal) | Cell signaling Technology | Cat# 9102, RRID:AB_330744 | WB (1:1000) |
| antibody | Phospho-p44/42 MAPK (Erk1/2) (Thr202/Tyr204) (rabbit monoclonal) | Cell signaling Technology | Cat# 4370, RRID:AB_2315112 | WB (1:1000) |
| antibody | Phospho-p44/42 MAPK (Erk1/2) (Thr202/Tyr204) (Mouse monoclonal) | Cell signaling Technology | Cat# 9106, RRID:AB_331768 | WB (1:1000) |
| antibody | FAK (D5O7U) XP (rabbit monoclonal) | Cell signaling Technology | Cat# 71433, RRID:AB_2799801 | WB (1:1000) |
| antibody | Phospho-FAK (Tyr397) (D20B1) (rabbit monoclonal) | Cell signaling Technology | Cat# 8556, RRID:AB_10891442 | WB (1:1000) |
| antibody | Phospho-FAK (Tyr576/577) (rabbit polyclonal) | Cell signaling Technology | Cat# 3281, RRID:AB_331079 | WB (1:1000) |
| antibody | Phospho-FAK (S843) (Clone 743101) (mouse monoclonal) | R & D Systems | Cat# MAB7298, RRID:AB_10994616 | WB (1:1000) |
| antibody | Phospho-Src (Y419) (rabbit polyclonal) | R & D Systems | Cat# AF2685, RRID:AB_442167 | WB (1:500) |
| antibody | Vimentin (V9) (mouse monoclonal) | Thermo Fisher Scientific | Cat# MA5-11883, RRID:AB_10985392 | WB (1:1000) |
| antibody | E-Cadherin (24E10) (rabbit monoclonal) | Cell signaling Technology | cat# 3195, RRID:AB_2291471 | WB 1:1000 |
| antibody | Rabbit Anti-HA-Tag (clone C29F4) (rabbit monoclonal) | Cell signaling Technology | Cat# 3724, RRID:AB_1549585 | WB 1:1000 |
| antibody | anti-c-Raf (rabbit polyclonal) | Cell signaling Technology | Cat# 9422, RRID:AB_390808 | WB 1:1000 |
| antibody | alpha/beta-Tubulin (rabbit polyclonal) | Cell signaling Technology | Cat# 2148, RRID:AB_2288042 | WB 1:1000 |
| antibody | GAPDH (0411) (mouse monoclonal) | Santa Cruz Biotechnology | sc-47724, RRID:AB_627678 | WB 1:2000 |
| antibody | Cyclin D1 (92G2) (rabbit monoclonal) | Cell signaling Technology | Cat# 2978, RRID:AB_2259616 | WB 1:1000 |
| antibody | PARP (rabbit polyclonal) | Cell signaling Technology | Cat# 9542, RRID:AB_2160739 | WB 1:1000 |
| antibody | Cleaved PARP (Asp214) (rabbit polyclonal) | Cell signaling Technology | Cat# 9541, RRID:AB_331426 | WB 1:1000 |
| antibody | β-Catenin (D10A8) XP (rabbit monoclonal) | Cell signaling Technology | Cat# 8480, RRID:AB_11127855 | WB 1:1000 |

*Continued on next page*

*Continued*

| Reagent type (species) or resource | Designation | Source or reference | Identifiers | Additional information |
|---|---|---|---|---|
| antibody | EpCAM (VU1D9) (mouse monoclonal) | Cell signaling Technology | Cat# 2929, RRID:AB_2098657 | WB 1:1000 |
| antibody | CD56 (NCAM) (mouse monoclonal) | Cell signaling Technology | Cat# 3576, RRID:AB_2149540 | WB 1:1000 |
| antibody | anti-β-Actin (mouse monoclonal) | Sigma-Aldrich | Cat# A5441, RRID:AB_476744 | WB 1:1000 |
| antibody | anti-GFP (Rabbit Polyclonal) | GeneTex | Cat# GTX113617, RRID:AB_1950371 | WB 1:1000 |
| chemical compound, drug | Collagen type I (from Human placenta) | Sigma-Aldrich | C7774 | 10 μg/well |
| chemical compound, drug | Collagen type III (From human placenta) | Advanced BioMatrix | 5019 | 10 μg/well |
| chemical compound, drug | Collagen type IV (From Human Cell) | Sigma-Aldrich | C7521 | 10 μg/well |
| chemical compound, drug | Fibronectin from Human Plasma | Millipore | FC010 | 2 μg/well |
| chemical compound, drug | Laminin (From EHS murine sacroma) | Sigma-Aldrich | L2020 | 10 μg/well |
| chemical compound, drug | TC-I 15 (ITGA2 inhibitor) | TOCRIS | 4527 | 1–10 μM |
| chemical compound, drug | ATN-161 (ITGA5 inhibitor) | TOCRIS | 6058 | 1–10 μM |
| chemical compound, drug | Defactinib (VS-6063) (FAK inhibitor) | Selleckchem | S7654 | 10 μM |
| chemical compound, drug | FR180204 (ERK inhibitor) | Sigma-Aldrich | SML-0320 | 10 μM |
| chemical compound, drug | Liberase | Roche | 5401054001 | |
| commercial assay or kit | FITC-Annexin V Apoptosis Detection Kit I | BD Biosciences | 556547 | FACS (5 uL per test) |
| commercial assay or kit | AnnexinV-FITC | Biolegend | 640906 | FACS (5 uL per test) |
| commercial assay or kit | CellTracker CM-DiI Dye | ThermoFisher Scientific | C7000 | |
| commercial assay or kit | CellTrace Violet Dye | ThermoFisher Scientific | C34571 | |
| commercial assay or kit | CellTrace FarRed Dye | ThermoFisher Scientific | C34572 | |
| chemical compound, drug | ProLong Gold antifade reagent | Cell signaling Technology | 9071 | |

*Continued on next page*

*Continued*

| Reagent type (species) or resource | Designation | Source or reference | Identifiers | Additional information |
|---|---|---|---|---|
| chemical compound, drug | Radioimmunprecipitation assay buffer (RIPA) | Cell signaling Technology | 9806 | |
| chemical compound, drug | Thiazolyl Blue Tetrazolium Bromide (MTT) | Sigma-Aldrich | M2128 | |
| chemical compound, drug | Agarose, low gelling temperature | Sigma-Aldrich | A9414 | |
| chemical compound, drug | Bovine serum albumin | Sigma-Aldrich | A9418 | |
| chemical compound, drug | Phosphatase Inhibitor Cocktail 2 | Sigma-Aldrich | P5726 | |
| chemical compound, drug | Phosphatase Inhibitor Cocktail 3 | Sigma-Aldrich | P0044 | |
| chemical compound, drug | Poteinase Inhibitor Cocktails | Sigma-Aldrich | P8340 | |
| chemical compound, drug | Cell-dissociation solution non-enzymatic | Sigma-Aldrich | C5914 | |
| commercial assay or kit | pGEM-T Easy Vector System | Promega | A1360 | |
| chemical compound, drug | Sequencing-grade modified trypsin | Promega | V5111 | |
| Recombinant DNA reagent | pUltra | Addgene | RRID:Addgene_24129 | |
| recombinant DNA reagent | pUltra-Chili | Addgene | RRID:Addgene_48687 | |
| recombinant DNA reagent | pCMVR8.74 | Addgene | RRID:Addgene_22036 | |
| recombinant DNA reagent | pMD2.G | Addgene | RRID:Addgene_12259 | |
| recombinant DNA reagent | mCherry-Integrin-Alpha2-N-18 | Addgene | RRID:Addgene_55063 | |
| recombinant DNA reagent | pSpCas9(BB)−2A-GFP | Addgene | RRID:Addgene_48138 | |
| recombinant DNA reagent | pSpCas9(BB)−2A-puro | Addgene | RRID:Addgene_48139 | |
| recombinant DNA reagent | lentiCRISPRv2 | Addgene | RRID:Addgene_52961 | |
| recombinant DNA reagent | pCMV3-ITGA2-HA | Sino Biological | HG13024-CY | |
| recombinant DNA reagent | pUltra-Chili-ITGA2-HA | this paper | ITGA2 overexpression | |
| recombinant DNA reagent | pUltra-ITGA2-HA | this paper | ITGA2 overexpression | |
| recombinant DNA reagent | lentiCRISPRv2-ITGA2 | this paper | ΔITGA2 | |

*Continued on next page*

*Continued*

| Reagent type (species) or resource | Designation | Source or reference | Identifiers | Additional information |
|---|---|---|---|---|
| recombinant DNA reagent | pSpCas9(BB)−2A-GFP _ITGA2-sgRNA1 | this paper | *ΔITGA2* | GGTTCAGCTACTGAGCTCTG |
| recombinant DNA reagent | pSpCas9(BB)−2A-GFP _ITGA2-sgRNA2 | this paper | *ΔITGA2* | GAAGGTCTACAGCTCTAGAA |
| recombinant DNA reagent | pSpCas9(BB)−2A-GFP _ITGA5-sgRNA1 | this paper | *ΔITGA5* | GTCCCGAGGAAGCAGAGCTG |
| recombinant DNA reagent | pSpCas9(BB)−2A-GFP _ITGA5-sgRNA3 | this paper | *ΔITGA5* | GCTAGGATGATGATCCACAG |
| sequence-based reagent | *ITGA2-gt-for* | this paper | PCR primers | ACGCCATCATGAGCAAGTCT |
| sequence-based reagent | *ITGA2-gt-rev* | this paper | PCR primers | GCAGCCGTGGTCTAAAAGGA |
| cell line (*Homo sapiens*) | IGROV-1 | ATCC | RRID:CVCL_1304 | Ovarian cancer cell lines |
| cell line (*H. sapiens*) | ES-2 | ATCC | RRID:CVCL_CZ94 | Ovarian cancer cell lines |
| cell line (*H. sapiens*) | OVCAR-3 | ATCC | RRID:CVCL_0465 | Ovarian cancer cell lines |
| cell line (*H. sapiens*) | OVCAR-4 | Charles River Laboratories | RRID:CVCL_1627 | Ovarian cancer cell lines |
| cell line (*H. sapiens*) | OVCAR-5 | Charles River Laboratories | RRID:CVCL_1628 | Ovarian cancer cell lines |
| cell line (*H. sapiens*) | OVCAR-8 | Charles River Laboratories | RRID:CVCL_1629 | Ovarian cancer cell lines |
| cell line (*H. sapiens*) | BG1 | Charles River Laboratories | RRID:CVCL_6570 | Ovarian cancer cell lines |
| cell line (*H. sapiens*) | OAW42 | ECACC | RRID:CVCL_1615 | Ovarian cancer cell lines |
| cell line (*H. sapiens*) | Kuramochi | JCRB | RRID:CVCL_1345 | Ovarian cancer ell lines |
| cell line (*H. sapiens*) | SK-OV-3 | ATCC | RRID:CVCL_0532 | Ovarian cancer cell lines |
| cell line (*H. sapiens*) | A2780 | ATCC | RRID:CVCL_0134 | Ovarian cancer cell lines |
| cell line (*H. sapiens*) | TOV-112D | ATCC | RRID:CVCL_3612 | Ovarian cancer cell lines |
| cell line (*H. sapiens*) | MeT-5A | ATCC | Cat# CRL-9444 RRID:CVCL_3749 | Human mesothelial cell line |

## Chemicals, reagents, and antibodies

The detailed list of primary antibodies, chemicals, and reagents using in biochemical or functional analyses are summarized in Key Resource Table.

## Cell culture

A total number of twelve different ovarian cancer cell lines were purchased via ATCC and Oncotest GmbH (now Charles River LaboratoriesInc) maintained in-house in RPMI-1640 supplemented with 10% fetal bovine serum (Sigma-Aldrich), 100 U/mL penicillin, and 0.1 mg/mL streptomycin unless stated differently (Key Resource Table). All the cell lines were regularly tested for mycoplasma contamination and authenticated using short tandem repeat STR profiling (Microsynth, Switzerland). The human mesothelial cell line MeT-5A was purchased from ATCC #CRL-9444 and cultured in Medium 199 supplemented with trace elements and additional growth factors (3.3 nM epidermal growth

factor, 400 nM hydrocortisone (Sigma-Aldrich), 870 nM bovine insulin (Sigma-Aldrich) and 20 nM HEPES). All cell lines were cultured at 37˚C in a 95% humidified atmosphere containing 5% $CO_2$. Patient ascites-derived tumor cells (ATCs) were cultured in DMEM/F-12 supplemented with 20% FBS, 100 U/mL penicillin, 0.1 mg/mL streptomycin, and 0.25 µg/mL amphotericin B.

## CRISPR-Cas9 sgRNA design and construction

Single guided RNAs (sgRNA) targeting exons 29–30 of *ITGA2* were designed using the web tool of UCSC Genome browser (https://genome.ucsc.edu/) (2). sgRNA1: GGTTCAGCTACTGAGCTCTG and sgRNA2: GAAGGTCTACAGCTCTAGAA respectively, were selected for gene editing of the transmembrane domain of *ITGA2*. Single guided RNAs (sgRNA) targeting exons 26–29 of ITGA5 were sgRNA1: GTCCCGAGGAAGCAGAGCTG and sgRNA3: GCTAGGATGATGATCCACAG. Oligo pairs encoding 20nt targeted sequences with overhangs (both 5' and 3') from *BbsI* restriction site were annealed and cloned into either pSpCas9(BB)−2A-GFP (addgene, #PX458) or pSpCas9(BB)−2A-puro (addgene, #PX459). Constructs were transformed into DH5alpha *E. coli* strains and confirmed by Sanger DNA sequencing using U6-F primer. Additionally, the same *ITGA2* oligos were cloned into lentiCRISPR v2 (addgene #52961).

## Generation of ∆*ITGA2* cell lines

EOC cell lines (IGROV1 and SKOV3) were grown in 6-well plates (5 × $10^5$ cells/well) for 24 hr and transiently transfected using TransIT-X2 transfection reagent (Mirus-Bio, Madison USA) with 3 µg of pair sgRNAs containing donor plasmids to generate homozygous *ITGA2* deletions. 72 hr after transfection, single cell sorting was performed on a BD FACS Aria Cell Sorter (BD Bioscience) sorting for single DAPI$^-$ and GFP$^+$ cells into 96-well flat-bottom plates with pre-warmed RPMI containing 10% FBS. Single cell clones were isolated and characterized by genotyping PCR, DNA sequencing, and immunoblotting. For generation of ∆*ITGA2* cell lines (OVCAR3, OAW42, and ES2), parental cells were transduced by lentiCRISPR v2-ITGA2 virus containing medium and selection by 2 µg/mL of puromycin for 2 weeks. The expression of ITGA2 was confirmed by western blot, flow cytometry, and immunofluorescence.

## Genotyping and DNA sequencing

Selected clones were characterized to identify homozygous knockout by using two independent PCR primer pairs ITGA2-gt-F: ACGCCATCATGAGCAAGTCT, ITGA2-gt-R: GCAGCCGTGGTC TAAAAGGA. PCR was performed using MyTaq (Bioline), 300 nM primer, and 50 ng genomic DNA. PCR conditions were 95˚C for three mins, then 35 cycles of 95˚C for 15 s, 58˚C for 20 s, 72˚C for 70 s, finished with 1 cycle at 72˚C for five mins. Amplicons were visualized on a 1.5% agarose gel. PCR products corresponding to the *ITGA2* genomic deletion regions were purified and cloned into the pGEM-T Easy Vector System (Promega) according to the manufacturer's instruction and sequenced using T7 and SP6 primer by Sanger DNA sequencing (Microsynth, Switzerland).

## Generation of ITGA2 overexpression construct and lentiviral transduction

The ITGA2 open reading frame was amplified using 2U of Pfu DNA polymerase (Promega), 1x Pfu polymerase buffer, 300 nM primers (ITGA2-XbaI_F: gaatctagaATGGGGCCAGAACGGACA and ITGA2-Nhe_R: caagctagcGCTACTGAGCTCTGTGGT), 30 ng cDNA template (Integrin α2-N-18; #addgene 55063), 200 µM dNTPs under following conditions: 95˚C for 1 min followed by 30 cycles of 95˚C for 30 s, 55˚C for 30 s, 72˚C for eight mins, and finished with 1 cycle at 72˚C for five mins. Amplicons were visualized on 1% agarose gel and purified by Wizard SV gel and PCR Clean-Up System (Promega) and cloning into C-terminal HA tagged pUltra (addgene#24129) or pUltra-Chili (addgene #48687) bicistronic expression vectors via XbaI/NheI cloning. The ligation products were transformed into Stbl3 *E. coli* competent cells, and the plasmids were purified and further verified by Sanger DNA sequencing (Microsynth, Switzerland). For preparation of lentiviral particles, HEK293T cells were seeded at around 50% confluency in a T75 flask one day before transfection. 4 µg of plasmid pUltra-ITGA2-HA (or pUltra-Chili-ITGA2-HA) and 2 µg of pMD2.G (Addgene #12259) and 2 µg of pCMVR8.74 (Addgene #22036) were co-transfected using 24 µL of jetPEI reagent in 1 mL of 150 mM NaCl solution (Polyplus-transfection, Chemie Brunschwig AG, Switzerland). Medium

was changed 24 hr after transfection. Virus supernatant was collected 48 hr later and filtered with a 0.45 μm polyvinylidene fluoride filter (Millipore). 3 mL of lentivirus-containing medium was used to transduce low-*ITGA2* expression cells in a T25 flask and sorted after three passages by GFP+ or dTomato+ population.

## Cell proliferation (proliferation index)

To identify the proliferation rate cells were seeded at a density of 10000 cells per well in 96-well standard cell culture plates, 2 μg/well fibronectin or 10 μg/well collagen-coated NuncUNC-Maxisorp plate. Cells were incubated for up to 96 hr. At each time point MTT dye (Sigma-Aldrich) was added at a final concentration of 500 μg/mL and incubated for 3 hr. After removal of supernatant, 200 μL of DMSO was added to dissolve the crystals. The optical density (OD, absorbance at 540 nm) was measured with a Synergy H1 Hybrid Reader (Biotek, Basel, Switzerland). Proliferation index was calculated as OD 540 nm at day four divided by day 2.

## Colony formation assay

Single cells were harvested and 1000 cells seeded per well of a 6-well tissue culture plate or precoated with collagen I 100 μg/mL at 4°C overnight. The standard culture medium was replaced every 3 days. 7 days after seeding, cells were fixed with 4% paraformaldehyde for 15 mins at room temperature and stained using 0.1% crystal violet. Images were acquired with Gel Doc XR+ and analyzed using ImageJ software.

## Soft agar spheroid formation assay

$10^4$ single cells were harvested and grown in culture medium containing 0.6% low gelling temperature base agarose (Sigma-Aldrich A9414) and 0.4% top agarose. Culture medium was added every 2–3 days. After 10–14 days of culture, colonies were fixed, stained and images acquired with Olympus IX81 microscopy.

## Cell migration assay

Cells were grown in serum-free medium for 24 hr and ~$1\times10^5$ cells were seeded into the upper chamber of a 12-well hanging inserts with a pore size of 8 μm (Millicell, Millipore). Cells were incubated at 37°C for 18 hr allowing cells to migrate toward the chemo-attractant (RPMI medium containing 10% FBS). After incubation, medium in the interior part of the insert was removed and the insert immersed in 0.1% crystal violet/4% paraformaldehyde solution for 20 mins. The insert was intensively washed and non-migrated cell were removed from the interior of the insert using a cotton-tip swab. Images were acquired with Olympus IX81 and the number of migrated cells were counted.

## Anoikis assay

To evaluate anchorage-independent cell growth, cells were grown in a 96-well ultra-low attachment plate (Corning Costar) for 3–5 days. Cells were then harvested, washed, and gently dissociated with 0.1% trypsin and stained with a FITC-Annexin V Apoptosis Detection kit (BD Biosciences) according to the manufacturer's instructions. The percentage of DAPI and FITC positive cells was analyzed by flow cytometry using CytoFLEX (Beckman Coulter).

## Cell adhesion assay

To evaluate the adhesion properties of cancer cells to ECM proteins, a 96-well Nunc Maxisorp flat-bottom plate (Invitrogen) were coated overnight at 4°C using 10 μg/well human type I, III, IV collagen, fibronectin, and Engelbreth-Holm-Swarm tumor (EHS)-derived laminin, which were prepared according to manufacturer's instructions. The plates were first washed with serum-free RPMI-1640 medium and 0.1% (w/v) BSA to remove unbound proteins, then blocked with serum-free RPMI-1640 medium and 0.5% (w/v) BSA for 45 mins at 37°C. EOC cell lines and ascites-derived tumor cells (under passage 0 to 1) were collected and resuspended in serum-free medium. A total of $5 \times 10^4$ cells were seeded in each well in triplicates and incubated for 20 mins at 37°C for adhesion. Non-adherent cells were removed by washing twice with serum-free medium with 0.1% (w/v) BSA. Adherent cells were fixed with 4% paraformaldehyde solution for 15 min, and stained with 0.05% (w/v)

crystal violet for 15 mins. Images were acquired using by widefield microscope Olympus IX81, and the number of adherent cells was analyzed using ImageJ software.

## Flow cytometry

Cell-surface integrin expression was analyzed by flow cytometry (CytoFLEX, Beckman Coulter) after antibody labeling. Sub-confluent cancer cell lines or primary ATCs were harvested, washed, and dissociated using 1X non-enzymatic dissociation buffer (Sigma-Aldrich). Cells were incubated with the following fluorescence-labeled antibodies: Alexa Fluor647 mouse anti-human ITGA2/CD49b (BD Bioscience, 1:100), FITC-mouse anti-human ITGA2/CD49b (1:100), Alexa Fluor647 mouse anti-human ITGA5/CD49e (1:100), BV510-mouse anti-human ITGB1/CD29(1:100) at 4°C for 1 hr. Matching isotype monoclonal antibodies conjugated to FITC or Alexa Fluor647 were used as controls (BD Bioscience). All investigated cell lines were gated individually to exclude debris, doublets, or DAPI-staining (BD Bioscience 0.1 µg/mL) to exclude dead cells. Data analysis was performed using FlowJo v10 BD (Becton Dickinson).

## Immunohistochemistry

Formalin-fixed, paraffin-embedded tissue samples were sectioned with a standard microtome at 3- to 5 µm thickness. After deparaffinization and rehydration, heat-induced (98°C) antigen retrieval was performed in 10 mM sodium citrate buffer (pH 6.0) at a sub-boiling temperature for 10 mins. The slides were incubated with hydrogen peroxide 3% (v/v) for 10 min, washed and blocked with 5% FBS in TBST for 1 hr at room temperature. Next, slides were incubated with primary antibodies anti-COL1A1 (1:100) (Sigma-Aldrich) and anti-ITGA2 (1:500) (Abcam) at 4°C overnight. Primary antibodies were detected using a SignalStain Boost IHC anti-rabbit HRP Reagent (Cell signaling Technology). The signal was visualized using a diaminobenzidine substrate kit (DAB, Thermo Fisher Scientific) according to the manufacturer's instructions and nuclei were counterstained with hematoxylin. Immunostaining was scored by the weighted average score (intensity: 0–3, coloring: 0–100% of ITGA2 expression) by two trained scientists independently and discrepancies were resolved by consensus.

## Immunofluorescence staining

Ovarian cancer cells were grown on an 8-well tissue culture chamber slides (Sarstedt, Switzerland), fixed with 4% paraformaldehyde, permeabilized with 0.3% Triton X100, and blocked with 5% (w/v) FBS (Sigma-Aldrich), 1% bovine serum albumin (BSA) fraction V, and 0.1% TritonX-100 containing PBS for 1 hr at room temperature. Cells were then stained with anti-integrin α2, anti-integrin β1, or anti-E-cadherin antibodies for overnight at 4°C. Following extensive washing, corresponding secondary antibodies were added to each chamber and incubated for 2 hr. Cells were washed with PBS containing 0.1% Tween 20 and incubated and counterstained with ProLong Gold antifade reagent with DAPI (Cell Signaling Technology). Fluorescence images were taken by a Zeiss LSM 710 confocal microscope (Zeiss, Feldbach, Switzerland). For patient-derived primary cells, 100–200 µL ascites cell suspension was collected using cytospin for 5 min at 400 g on a glass slide. Cells were then fixed with 4% paraformaldehyde, permeabilized, blocked, and stained as mentioned above.

## Immunoblotting

Cells were lysed in 1X radioimmunprecipitation assay buffer (RIPA, Cell Signaling Technology) containing proteinase inhibitor cocktails (Sigma-Aldrich). Lysates were clarified by centrifugation at 18,000 g for 15 min at 4°C. Clarified lysates were boiled in 1x sample buffer (50 mM Tris-HCl, 1% SDS, 100 mM DTT and 10% glycerol) at 95°C for 5 min and resolved by SDS-PAGE. Proteins were then transferred to a polyvinylidene difluoride (PVDF) membrane (BioRad) and blocked with 5% (w/v) bovine serum albumin in TBST (20 mM Tris-Base, 150 mM NaCl, pH 7.8, 0.1% Tween 20) for 1 hr at room temperature. The membrane was incubated with one of the listed primary antibodies diluted in 5% (w/v) BSA in TBST overnight at 4°C. After extensive washing in TBST, the membrane was incubated with corresponding HRP-conjugated secondary antibodies (1:10000, Cell Signaling Technology) for 3 hr at room temperature. Finally, the membrane was developed using the Super Signal West Dura Extended Duration Substrate (Thermo Fisher Scientific) for detection of HRP. western blot results were visualized by Gel Doc XR+ (BioRad) and analyzed by Image Lab software (BioRad).

## Cancer cell to mesothelial adhesion assay

A 24-well plate was coated with 5 µg/mL collagen I and mesothelial cells MeT-5A were cultured until full confluency was reached. WT cancer cells were labeled with CellTrace (Thermo Fisher Scientific) Far Red (Ex/Em 630/661 nm) and ΔITGA2 cancer cells with CellTrace Violet dye (Ex/Em 405/450 nm) for 20 mins at 37°C according to the manufacturer's instruction. WT and ΔITGA2 cells were equally mixed and added on top of the mesothelial cell layer. The adhesion of cancer cells was measured over time by washing the plate twice with PBS followed by detachment of all the adherent cells using trypsin for subsequent flow cytometry analysis using CytoFLEX (Beckman Coulter). The cell adhesion ratio was calculated by dividing the percentage of WT cells to the ΔITGA2 cells.

## Mesothelial clearance assay

The mesothelial clearance assay was modified from *Iwanicki et al., 2011*. In brief, 100–200 EOC cells which stably expressed red fluorescent dTomato protein were cultured in a 96-well ultra-low attachment plate for 1–2 days in order to generate multicellular spheroids. GFP-expressing mesothelial cells (MeT-5A) were seeded in a 6-well plate at 37°C overnight to allow the mesothelial cells to form a confluent monolayer. The cancer cell spheroids (10–20/per well) were collected and gently added on top of the mesothelial monolayer. To monitor the intercalation process in a real-time quantitative manner, we performed the time-lapse imaging by installing a 12-well cell culture plate on a motorized stage capable of imaging multiple positions. GFP, RFP, and bright field images were acquired every 30 mins for 24 hr using the widefield fluorescence microscope Olympus IX81 with a custom built-in incubation chamber with temperature (37°C) and $CO_2$ control. To quantify mesothelial clearance, the non-fluorescent area (black hole) in the GFP mesothelial monolayer was measured over time and divided by the initial area of the cancer spheroid (at time zero). At least 10 spheroids were quantified for each cell line and all experiments were performed in triplicates.

## Zebrafish xenograft model

All zebrafish experiments and husbandry were in compliance with the Swiss Animal Protection Ordinance and approved by the Kantonales Veterinaeramt Basel-Stadt. The xenotransplantation experiments were performed as described previously (*Jacob et al., 2018*). In brief, WT and ΔITGA2 cancer cells were labeled with the fluorescent CellTracker CM-DiI (Life Technologies). Zebrafish embryos were anesthetized in 0.4% tricaine (Sigma-Aldrich) at 2 day post fertilization (dpf) and 100–150 human EOC cells were microinjected into the vessel free area of the yolk or the zebrafish common cardinal vein (Duct of Cuvier) of a transgenic Tg(*kdrl:eGFP*) line, respectively. Fish harboring red cells were incubated at 35°C. Five days after transplantation, embryos were screened microscopically for tumor cell cluster formation and extravasation using a Leica TCS SP5 confocal microscope. Fish were furthermore dissociated into single cells as described elsewhere (*Jacob et al., 2018*), and cells were analyzed on a CytoFELX for DAPI negative and CM-DiI-positive cells. For each experiment at least five fish for each condition were analyzed in multiple biological replicates.

## Intraperitoneal xenograft model

All animal experiments were performed in accordance with the European Guidelines for the Care and Use of Laboratory Animals, Directive 2010/63/UE, Portuguese National Regulation published in 2013 (Decreto-Lei n.8 113/2013 de seven de Agosto) and approved by the local Ethics Committee of the Institute for Research and Innovation in Health (i3S) (Porto, Portugal). Project identification code (0421/000/000/2017, date (24/05/17)). The authors involved in these experiments have an accreditation for animal research given from the Portuguese Veterinary Board (ministerial Directive 1005/92). NIH(S) II: nu/nu mice were generated under IPATIMUP supervision. To generate intraperitoneal xenografts, $4 \times 10^6$ of WT and ΔITGA2 SKOV3 cells were resuspended in 200 µL of PBS and intraperitoneally injected, using 25 gauge needles, in female NIH(S) II: nu/nu mice with 6–8 weeks of age. Eight weeks after injection, mice were humanely euthanized (with anesthesia followed by cervical dislocation) and peritoneal and pleural cavities were carefully inspected. Animal organs were harvested for histological processing. H & E staining from all tissue blocks were examined under the microscope to evaluate tumor localization, growth, and invasion. Six mice per group were used in a total of two independent experiments.

## Time-lapse imaging and cell tracking of cancer cell on collagen

WT and ΔITGA2 or ITGA2-overexpressed (OE) cancer cells were detached from a T-75 flask by incubating with 2 mL of cell-dissociation reagent (Sigma-Aldrich) for 10–15 mins followed by addition of 3 mL of fresh medium. 125,000 cells in 1.2 mL of fresh medium were added onto a collagen type I coated 4-well Lab-Tek II Chamber Slide using bioprinting technique (3D Discovery, regenHU Ltd. Switzerland). Cells were allowed to adhere for 30 mins at 37°C and 5% $CO_2$ prior to live cell imaging. Next, cells in each well (two positions) were monitored using the time-lapse imaging option available on a Zeiss LSM 710 fluorescence confocal microscope (Zeiss, Germany) equipped with the live chamber module. The samples were excited using a 561 nm laser in z-stack mode (2 µm slice thickness, 20x magnification) for every five mins for a total duration of 6 hr. Images were preprocessed using the maximum intensity projection tool available in Zen 2011 software (Zeiss, Germany). Surface area calculation was determined by using image processing in Matlab (MathWorks, US) consisting of: (1) intensity thresholding, (2) binarization, and (3) pixel area calculation. Surface area ratio was obtained by comparing the obtained value with the initial cell area at t = 0 hr. Cell velocity and displacement were measured by using the TrackMate plugin available in Fiji (NIH, US) (*Tinevez et al., 2017*). A minimum of 300 cells of each sample was subjected to the surface area and cell tracking analysis. All experiments were performed in triplicates.

## Scanning Electron Microscope

Sterile glass cover slips were coated with collagen type I using bioprinting technology and $2 \times 10^4$ SKOV3 WT and ΔITGA2 cells were added. Cells were allowed to adhere for 1, 6, and 24 hr and then fixed overnight at 4°C using 2.5% glutaraldehyde in 0.03 M $K_2HPO_4/KH_2PO_4$ buffer and 2% paraformaldehyde in PBS. Samples were washed with PBS, dehydrated using series of ethanol dilutions (20–100%; five mins per dilution), and dried with hexamethyldisilazane for 15 mins. Samples were sputtered-coated with 3 nm gold and analyzed using a scanning electron microscopy (Tescan Mira3 LM FE, USA). All experiments were performed in duplicates.

## Proteomic analysis of omental metastasis and normal omentum

Each fresh piece of omentum tissue (approximately 50 mg, n = 5) was immediately grinded by a tissue homogenizer with lysis buffer containing protease inhibitor cocktails (Sigma-Aldrich), 0.1M ammonium bicarbonate buffer and 8M urea on ice. Tissue extracts were then sonicated using Bioruptor (Diagenode, Belgium) with the standard program (30 s on, 30 s off for 10 cycles) followed by incubation at 37°C for 60 mins at 1400 rpm. Protein concentration was measured by standard BCA assays according to the manufacturer's instruction (Thermo Fisher Scientific). 50 µg of total protein was reduced with 5 mM TCEP for 60 mins at 37°C, alkylated with 10 mM chloroacetamide for 30 mins at 37°C, and digested with sequencing-grade modified trypsin (1/50, w/w; Promega, Madison, Wisconsin) overnight at 37°C. Digested peptides were cleaned up using iST cartridges (PreOmics, Munich) according to the manufacturer's instruction. Samples were resuspended in 0.1% formic acid by sonication and subjected to LC–MS/MS analysis using an Orbitrap Fusion Lumos Tribrid Mass Spectrometer fitted with an EASY-nLC 1200 (Thermo Fisher Scientific). Peptides were resolved using a RP-HPLC column (75 µm × 36 cm) packed in-house with C18 resin (ReproSil-Pur C18–AQ, 1.9 µm resin; Dr. Maisch GmbH) at a flow rate of 200 nL/min using solvent A (0.1% formic acid in water) and solvent B (0.1% formic acid in 80% of LC–MS grade acetonitrile) as mobile phases.

The mass spectrometer was operated in a data-dependent analysis (DDA) mode with a cycle time of 3 s between MS1 scans. Each MS1 scan was acquired in the Orbitrap at a resolution of 120,000 FWHM (at 200 m/z) and a scan range from 375 to 1600 m/z followed by MS2 scans of the most intense precursors in the linear ion trap at 'Rapid' scan rate with isolation with of the quadrupole set to 1.4 m/z. Maximum ion injection time was set to 50 ms (MS1) and 35 ms (MS2) with an AGC target of 1e6 and 1e4, respectively. Only peptides with charge state 2–5 were included in the analysis. Monoisotopic precursor selection (MIPS) was set to peptide, and the Intensity Threshold was set to 5e3. Peptides were fragmented by HCD (Higher-energy collisional dissociation, normalized collision energy 35%). The acquired raw-files were converted to mgf format using MS Convert and searched using MASCOT for a human database from Uniprot on 20190129. Mass tolerance of 10 ppm (precursor) and 0.6 Da (fragments). The database search results were filtered using the ion score to set the

false discovery rate (FDR) to 1% on both peptide and protein levels with Scaffold 4.0 proteomics software.

## Phosphoproteomics and data analysis

WT and ΔITGA2 cancer cells (IGROV1, SKOV3, and OVCAR3) were starved in medium without FBS for 6 hr and stimulated with 10% FBS for 30 min. After mechanically detached, cells were washed and lysed in 8M Urea, 0.1M ammonium bicarbonate in presence of phosphatase inhibitors (Sigma-Aldrich) using ultra-sonication (Bioruptor, Diagenode, Belgium, 10 cycles, 30 s on/off). Proteins were reduced, alkylated, and digested as mentioned above. After acidification using 5% TFA, peptides were desalted on C18 reversed-phase spin columns according to the manufacturer's instructions (Macrospin, Harvard Apparatus), dried under vacuum and stored at −20°C until further use. Peptide samples were enriched for phosphorylated peptides using Fe(III)-IMAC cartridges on an AssayMAP Bravo platform. Chromatographic separation of peptides was carried out using an EASY nano-LC 1200 system (Thermo Fisher Scientific), equipped with a heated RP-HPLC column (75 μm x 36 cm) packed in-house with 1.9 μm C18 resin (ReproSil-Pur C18–AQ, 1.9 μm resin; Dr. Maisch GmbH). Aliquots of 0.75 μg total peptides were analyzed per LC-MS/MS run at a flow rate of 200 nL/min. Mass spectrometry analysis was performed using an Orbitrap Fusion Lumos Tribrid Mass Spectrometer equipped with a nanoelectrospray ion source (Thermo Fisher Scientific). The acquired raw-files were imported into the Progenesis QI software (v2.0, Nonlinear Dynamics Limited). Quantitative analysis results from label-free quantification were processed using the SafeQuant R package v.2.3.2. (https://github.com/eahrne/SafeQuant/) to obtain peptide relative abundances. This analysis included global data normalization by equalizing the total peak/reporter areas across all LC-MS runs, data imputation using the knn algorithm, summation of peak areas followed by calculation of peptide abundance ratios. Only isoform-specific peptide ion signals were considered for quantification. The summarized peptide expression values were used for statistical testing of differentially abundant peptides. Here, empirical Bayes moderated t-Tests were applied using limma package (http://bioconductor.org/packages/release/bioc/html/limma.html). Heatmap analysis of KEGG hsa04510:focal adhesion pathway phosphoproteins between and the KEGG pathview package were analyzed using Pathview: An R/Bioconductor package for pathway-based data integration and visualization (*Luo and Brouwer, 2013*; *Figure 7F* and *Figure 7—figure supplement 1A*).

## Access and bioinformatics analysis of publicly available datasets

Publicly available transcriptomic datasets of *curatedOvarianData* were downloaded from Gene Expression Omnibus (http://www.ncbi.nlm.nih.gov/geo/). The *curatedOvarianData* package provides a comprehensive resource of curated gene expression and clinical data for the development and validation of ovarian cancer prognostic models (*Ganzfried et al., 2013*). Statistical analysis of hazard ratio in the forest plot (*Figure 1B* and *Figure 1—figure supplement 1D*) were obtained through the use of the software R version 3.5.3 (www.R-project.org) and instruction from *curatedOvarianData* package (*Ganzfried et al., 2013*). p-values were calculate accordingly using fixed-effects model. Gene expression data for the Cancer Cell Line Encyclopedia (CCLE) were accessed through the cBioPortal using R (www.cbioportal.org) and cgdsr for querying the Cancer Genomics Data Server as described (*Barretina et al., 2012*; *Figure 2A and C*, *Figure 1—figure supplement 2*, and *Figure 5—figure supplement 1C*).

## Gene set variation analysis (GSVA) for collagen gene set

The RNA-seq data of normal tissues in GTEx (the Genotype-Tissue Expression Project, (dbGaP Accession phs000424.v8.p2) on 26/08/2019) and ovarian cancers in TCGA (The Cancer Genome Atlas) were downloaded from UCSC Xena Cancer Genomics Browser (https://xena.ucsc.edu/). To analyze the enrichment of common collagen genes in normal tissues and ovarian cancer tissues, the GSVA method (*Hänzelmann et al., 2013*) were performed to acquire the GSVA enrichment score of collagen gene set from each patient. Using a nonparametric approach, GSVA transforms a gene by sample matrix into a gene set by sample matrix, facilitating the identification of various functional related collagen genes. The box plot represented enrichment scores in selected normal tissues from GTEx RNA-seq dataset (*Figure 1C*). The scatter plots represented the correlation between ITGA2 expressions and collagen GSVA enrichment scores were generated using ggplot2 and ggpubr

package in R (*Figure 1—figure supplement 3D–E*). Pearson linear correlation was used to calculate the correlation coefficient of ITGA2 expression and collagen enrichment score.

## Reactome pathway enrichment analysis

Phosphoproteomics identified proteins (FDR < 1%) with at least one hypo-phosphorylated site (log2FC > 2 and p-value<0.01) or with at least one hyper-phosphorylated site (log2FC < −2 and p-value<0.01) in IGROV1 ΔITGA2 cells were selected for the pathway enrichment analysis. The ReactomePA (*Yu and He, 2016*) and clusterProfiler (*Yu et al., 2012*) R packages were applied to calculate the Reactome pathway enrichment and visualize the enrichment results as shown in (*Figure 7C and D*).

## Statistical analysis

All data including error bars are presented as mean ± SD in triplicates unless otherwise stated. Statistical calculations were performed using GraphPad Prism 8.0. Two experimental groups were compared by using unpaired Student's t-tests. Where more than two groups were compared, a one-way ANOVA with Bonferroni's correction was used. p-values < 0.05 were considered statistically significant (***, p<0.001, **p<0.01, *p<0.05).

## Acknowledgements

This work was supported by the Swiss National Foundation (Sinergia 171037 to V Heinzelmann-Schwarz and B Rothen-Rutishauser). Nachwuchsförderung Klinische Forschung was provided by the University of Basel (DMM2051) to YL Huang. FreeNovation 2016 was provided by Novartis to F Jacob. We are grateful to the Department of Biomedicine and Biozentrum, University of Basel for providing all the necessary support and for allowing us to work in the Flow Cytometry Facility (Emmanuel Traunecker, Lorenzo Raeli, and Telma Lopes), the Microscopy Facility (Michael Abanto and Beat Erne), and the Proteomics Core Facility (Thomas Klaus Christian Bock and Alexander Schmidt). We would also like to acknowledge Nicki Packer for her critical review of the manuscript. We thank Christian Beisel and Ulrike Menzel from the Genomics Facility Basel and Franziska Singer from NEXUS Clinical Bioinformatics at ETH Zurich for their support.

## Additional information

### Funding

| Funder | Grant reference number | Author |
|---|---|---|
| Swiss National Science Foundation | Sinergia 171037 | Barbara Rothen-Rutishauser Viola Heinzelmann-Schwarz |
| University of Basel | DMM2051 | Yen-Lin Huang |
| FreeNovation | FreeNovation 2016 | Francis Jacob |

The funders had no role in study design, data collection and interpretation, or the decision to submit the work for publication.

### Author contributions

Yen-Lin Huang, Conceptualization, Data curation, Formal analysis, Investigation, Methodology, Writing - original draft, Writing - review and editing; Ching-Yeu Liang, Data curation, Software, Formal analysis, Investigation, Methodology, Assisted with bioinformatic analysis; Danilo Ritz, Data curation, Investigation, Methodology, Assisted with proteomics and phosphoproteomic analysis; Ricardo Coelho, Data curation, Investigation, Methodology, Assisted with animal experiments; Dedy Septiadi, Manuela Estermann, Data curation, Investigation, Assisted with Time-lapse imaging and Scanning Electron Microscopy; Cécile Cumin, Mónica Núñez López, Diego Calabrese, Investigation; Natalie Rimmer, Investigation, Writing - review and editing, Assisted with animal experiments; Andreas Schötzau, Data curation, Software, Assisted with bioinformatic coding; André Fedier, Investigation, Project administration; Martina Konantz, Methodology, Assisted with animal experiments;

Tatjana Vlajnic, Resources, Investigation; Claudia Lengerke, Supervision; Leonor David, Barbara Rothen-Rutishauser, Conceptualization, Supervision; Francis Jacob, Conceptualization, Data curation, Supervision, Investigation, Methodology, Writing - review and editing; Viola Heinzelmann-Schwarz, Conceptualization, Resources, Supervision, Funding acquisition, Writing - review and editing

### Author ORCIDs
Yen-Lin Huang (iD) https://orcid.org/0000-0002-3678-3815
Ching-Yeu Liang (iD) https://orcid.org/0000-0002-9027-1853
Martina Konantz (iD) http://orcid.org/0000-0002-4319-3119
Francis Jacob (iD) https://orcid.org/0000-0002-0446-1942

### Ethics
Human subjects: The study was approved by the respective medical ethics committees: The Swiss Medical Ethical Committee (EKNZ 2015-436) and the Hunter New England Human Research Ethics Committee (HNEHREC 16/04/20/5.06). The application of patient-derived materials is summarized in Table S3.

Animal experimentation: All mice experiments were performed in accordance with the European Guidelines for the Care and Use of Laboratory Animals, Directive 2010/63/UE, Portuguese National Regulation published in 2013 (Decreto-Lei n.8 113/2013 de 7 de Agosto) and approved by the local Ethics Committee of the Institute for Research and Innovation in Health (i3S) (Porto, Portugal). Project identification code (0421/000/000/2017, date (24/05/17)). The authors involved in these experiments have an accreditation for animal research given from the Portuguese Veterinary Board (ministerial Directive 1005/92).

### Decision letter and Author response
Decision letter https://doi.org/10.7554/eLife.59442.sa1
Author response https://doi.org/10.7554/eLife.59442.sa2

# Additional files

### Supplementary files
• Supplementary file 1. Proteomic, phosphoproteomic analysis, and clinical information of patient samples used in this study. (a) Selected differentially expressed cellular adhesion and ECM-related proteins in omentum metastasis (T) vs normal omentum (O). (b) 69 significantly downregulated phosphoproteins shared among three ΔITGA2 EOC cell lines. (c) Selected significantly down and upregulated phosphopeptides in comparison between ΔITGA2 and WT IGROV1 cells. (d) Clinical and pathological information of patient samples used in this study (e) Clinical and pathological information of patient samples in tissue microarray (TMA).

• Transparent reporting form

### Data availability
All data generated or analysed during this study are included in the manuscript and supporting files. Source data files have been provided in supplementary files.

The following previously published datasets were used:

| Author(s) | Year | Dataset title | Dataset URL | Database and Identifier |
|---|---|---|---|---|
| Frederick BG | 2013 | curatedOvarianData | https://bcb.dfci.harvard.edu/ovariancancer | curatedOvarianData, ovariancancer |
| Genomic Data Commons | 2019 | GDC TCGA Ovarian Cancer | https://gdc.xenahubs.net/download/TCGA-OV.htseq_fpkm-uq.tsv.gz | TCGA-OV.htseq_fpkm, htseq_fpkm |
| Ucsc TOIL RNA-seq recompute | 2016 | GTEX; gene expression RNAseq | https://toil.xenahubs.net/download/gtex_RSEM_gene_tpm.gz | gtex_RSEM_gene_tpm, gtex_RSEM_gene_tpm.gz |

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
