## [Decision Letter]

**Acceptance summary:**

This article provides convincing evidence supporting an important role of interactions between collagens and integrin α 2 (ITGA2) in the seeding of ovarian cancer cells to the omentum during metastasis of ovarian cancers. The study integrates the analysis of clinical specimens with extensive functional assessment of the importance of ITGA2 versus other integrins in directing adhesion, and suggests potential directions for future therapeutic development.

**Decision letter after peer review:**

Thank you for submitting your article "Collagen-rich omentum is a premetastatic niche for integrin α2 mediated peritoneal metastasis" for consideration by *eLife*. Your article has been reviewed by three peer reviewers, one of whom is a member of our Board of Reviewing Editors, and the evaluation has been overseen by a Senior Editor. The reviewers have opted to remain anonymous.

As is customary in *eLife*, the reviewers have discussed their critiques with one another. What follows below is the Reviewing Editor's compilation of the essential and ancillary points provided by reviewers in their critiques and in their interaction post-review. Please aim to submit a revised version that addresses these concerns directly. Although we expect that you will address these comments in your response letter we also need to see the corresponding revision in the text of the manuscript. Some of the reviewers' comments may seem to be simple queries or challenges that do not prompt revisions to the text. Please keep in mind, however, that readers may have the same perspective as the reviewers. Therefore, it is essential that you attempt to amend or expand the text to clarify the narrative accordingly.

Summary:

In this paper, Huang et al. address the role of interactions between collagens and integrin α 2 (ITGA2) in the seeding of ovarian cancer cells to the omentum. Strengths of the study include the use of a lot of clinical samples, and the first demonstration of a dependence on integrin α 2 (ITGA2) for cell adhesion and migration in ovarian cancer. There is an enormous amount of carefully performed work, and a lot of comprehensive profiling. However, the study has some weaknesses that need to be addressed.

In this study, the authors begin by using proteomics to assess the expression of proteins in the normal omentum, versus omentum with an embedded metastatic ovarian cancer. This identified upregulation of a number of collagens, fibronectin, and matrix metalloproteases, among others. They found that a number of integrins were expressed similarly in both normal omentum and omentum bearing a metastasis, whereas, ITGA2 was expressed at higher levels in the metastatic setting. They showed that ITGA2 and CDH1 are expressed in more than half of spheroids among ascites tumor cells (ATCs). ITGA2 was knocked out in 5 distinct ovarian cancer cell lines of various cell types. This did not affect proliferation, but did reduce chemotaxis (10% FBS)-induced cell motility and tumor cluster formation, and increase anoikis, implying defects in attachment.

The authors directly demonstrate that knockdown of ITGA2 reduces adhesion to collagen but not fibronectin, and reduced cell spreading and formation of focal adhesions, and overexpression of ITGA2 increases adhesion. Analyzing patient samples, they show distinct paterns of expression of ITGA2 and ITGA5. They show that the ITGA2 inhibitor TC-I-15 blocks collagen adhesion even in ovarian cancer cells overexpressing both ITGA2 and ITGA5. They demonstrate that cell lines lacking ITGA2 have reduced adhesion to mesothelial cells, are less capable of mesothelial clearance, and form fewer tumors following IP injection. Finally, the authors perform proteomic and phosphoproteomic profiling of ITGA2 wt and mutant cells grown in the presence of collagen. They find loss of ITGA2 causes cells to acquire more epithelial features, and to downregulate the focal adhesion signaling cascade. They then show that treatment with a focal adhesion kinase (FAK) inhibitor defactinib reduces EOC growth, and identify a correlation with high ITGA2 expression and high activation of FAK and associated proteins (SRC, ERK) in omental metastases.

The manuscript contains some interesting data. Further understanding of the functional role of ITGA2 in OC is important, as the role of this protein has not been extensively explored, with only a few previous publications. Overall, the work employed appropriate methods to interrogate the effects of genetic loss, pharmacologic inhibition or overexpression of ITGA2 in OC cells, including use of multiple cell lines and human OC tumor/ascites specimens and generally rigorous analyses. The clinical correlation in Figures 1-2 are a nice clinically relevant start for the manuscript.

However, some parts of the manuscript seem more like a collection of phenotypic/descriptive data rather than experiments aimed and investigating the big question – the role of ITGA2-col axis in EOC metastasis. Findings are sometimes generalized across all cell lines when the data for individual cell lines show key significant differences. While the phosphoproteomic analysis is interesting, it only one cell line was subjected to analysis. FAK/MAPK signaling may play a major role in collagen-ITGA2 mediated signaling, but phosphoproteomic survey alone is insufficient to definitively demonstrate this is the primary mechanism, particularly if a single cell line was used.

The major limitation of the study is that there is little mechanistic insight. The authors could increase the impact by investigating at least some of the potential lead further: how does ITGA2 contributes to anoikis resistance in a ligand independent manner (as the a2 inhibitor had no effect on spheroid formation), why does loss of a2 induce epithelial features – is a2 contributing to anchorage independence and metastasis triggering loss through regulation of EMT/stemness features? What is the role of a2-collagen feed-forward signaling (Figure 7) to supporting a collagen rich metastatic niche? Suggestions are offered below to strengthen the data presented and further support the conclusions of the study.

Essential revisions:

1) Although ITGA5 is listed as non-significantly elevated in patient samples, it looks as if very few normal specimens were profiled for this integrin, compared to the others, for some reason. This is a particularly important issue to evaluate carefully, given several publications that have noted a specific and critical role for ITGA5 in promoting ovarian cancer metastasis (among others, PMID: 30710055, PMID: 23499550, PMID: 18381440), given the images provided in Figure 1E suggest the expression of this protein is particularly heterogeneous among tumors, and given the authors' data in Figure 5A actually suggests it is abundant. They also exclude ITGA5 from important subsequent analysis (e.g., database and direct cell line evaluation in Figure 2, 4, etc). This eliminates a critical benchmark; for example, the authors demonstrate using three cell lines in which ITGA5 is knocked out that binding to collagen is lost. If these three lines happen to be rare ones lacking ITGA5 expression, this could explain the dramatic phenotype. Information about ITGA5 should be added consistently.

2) In a survey of cell models, ITGA2 expression did not correlate with histotype or TP53 mutation status. This is curious, since serous ovarian cancers are the most aggressive and metastatic; one might expect to see higher levels in this cancer type. What is the author's explanation for this?

3) In Figure 5, the PIs use the ITGA2-specific inhibitor in cell lines overexpressing both ITGA2 and ITGA5 to demonstrate that inhibition of ITGA2 is sufficient to disrupt binding to collagen. However, they use this inhibitor at 40 μM. From the Tocris web page, "Potent α2β1 integrin inhibitor (IC50 values for the inhibition of human platelet adhesion to type I collagen are 12 and 715 nM for platelets under static conditions and under flow, respectively). Displays selectivity for α2β1 over αvβ3, α5β1, α6β1 and αIIbβ3 at concentrations exceeding 1000 nM." 40 μM is far in excess of these concentrations. The assay should be performed at appropriate concentrations; for instance, 1 μM. In addition, Tocris sells an α5 inhibitor, ATN161; this should be used in parallel assays, with the expectation it would not have an effect at concentrations where it is active for ITGA5 but not ITGA2.

4) Interestingly, from the data in Figure 5A show a partial although incomplete overlap between ITGA2 and ITGA5 expression. In Figure 5B, the authors show "a strong positive correlation between the percentage of ITGA2+ ATCs and the adhesion efficiency to collagen". A similar analysis should be done for ITGA5.

5) The importance of FAK signaling in EOC is well-established, and targeting of FAK is already being explored clinically in these cancers (for instance, PMID: 31478830), reducing the novelty of the last part of the study. Hence, bolstering conviction of an activity specific for ITGA2 is important. FAK is activated by multiple integrins, and its activation in response to ITGA5 is well-established. It would be important to benchmark ITGA2 and ITGA5 in some of the experiments in Figure 7. For instance, do the cell lines analyzed by phosphoproteomic analysis express or not express ITGA5? Do the metastatic cells in the omentum with high ITGA2 also have ITGA5? Are there examples with high ITGA2 but not ITGA5? These seemed to be relatively rare based on data shown in Figure 5A. These data are extremely important for assessing the specificity of the signaling, and in understanding the relevant clinical populations likely to respond to FAK inhibitors.

6) It is actually interesting that expression of ITGA2 and ITGA5 seem to correlate (based on data provided). Do the authors have any thoughts as to whether this correlation is real, and if so, what is causing it?

7) The data in Figure 5-7 are the most exciting. Are the metastatic foci rich in collagen? Does FAK inhibition have an effect on this? FAK is a key regulator of anoikis but also functions downstream of integrin to regulate adhesion turnover etc. Is the FAKi effect (Figure 7G-H) limited only to inhibiting sphere growth or does it also influence the mesothelium clearance/collagen interaction? The authors should clarify their description and interpretation of the elevated expression of collagens in omentum bearing metastases. Certainly, the fact that the omentum changes expression of collagens over time is well known, as is the relationship of this to onset of aging-related pathologies (one recent review – PMID: 29996539). Some of these changes may "prime the soil" for seeding by ovarian cancers expressing cognate integrins. However, the analysis of phosphoproteomic and TCGA data at the end of the paper seems to suggest the collagens are being produced by the tumor cells themselves – which would lead to a different model.

8) Although ITGA2 has not been as heavily investigated as other integrins for a role in EOC, one study found it to be upregulated in ovarian cancer (PMID: 31443152), and a more recent one showed it contributed to paclitaxel resistance in ovarian cancer by activating AKT/paclitaxel (PMID: 32202508). The discussion should discuss these works in greater detail; in particular, it would be useful to place the current work in the context of AKT pathway activation, given this is a major regulator of anoikis, and engages in cross-talk with FAK, SRC, and ERK.

9) Authors state that ΔITG2A cells "maintained adhesion to laminin and fibronectin (Figure 4, A-B)", but the data showed variability across cell lines, suggesting this was not generalizable. While true for SKOV3 cells, other cell lines showed significant differences in one or the other substrate. This variability in effect on FN or LN binding is also observed in experiments utilizing the ITGA2 inhibitor TC-I-15 (shown in Figure 5). The text describing the findings in the context of FN and LN should be modified.

10) Although the authors tested the effects of TC-I-15 on AS20 cells and showed no significant change, use of these cells only does not fully address the specificity of TC-I-15 for ITGA2 as ITGA5 protein levels in these cells are mid-range. Testing of cells that exhibit ITGA2 low ITGA5 high would further support the selectivity of drug effect for ITGA2.

11) The finding that TC-I-15 has no effect on spheroid formation seems at odds with the prediction that ITGA2 is a key contributor to – if not essential for – sphere formation and escape from detachment-induced anoikis as shown in Figure 3D and E. The authors show OE of ITGA2 results in increased anchorage independent colony formation. Was the effect of deletion of ITGA2 on anchorage independent colony size and number tested (WT vs ΔITG2A cells)?

12) in vivo analysis of SKOV3 WT and ΔITGA2 cells injected in nude mice showed significant reduction of total tumor implants in the ΔITGA2 xenografts. In Figure 3D, the authors show that SKOV3 ΔITGA2 cells grown on non-adherent plates exhibited a high degree of cell death (46.6%) as compared to SKOV3 control cells (12.9%). Was the potential difference in viability of the injected cells addressed to exclude the possibility that the reduction in tumor growth was due to significantly reduced viability of ΔITGA2 resulting in fewer cells to capable of growth? This is an essential point to clarify before one can conclude that ITGA2 is promoting dissemination and invasion of collagen rich omentum.

13) Phosphoproteomic analysis led the authors to conclude that ITGA2-collagen interaction fosters cancer cell survival, cell migration and anoikis resistance through the FAK/MAPK signaling axis. The only cell line mentioned in this analysis is IGROV-1. The implications of the findings would be substantially strengthened by analysis of additional cell lines with comparative analysis across lines to substantiate key findings. Functional assays would lend further support to the conclusion that FAK/MAPK signaling axis is the primary mechanism.

14) The findings of this study should include deeper discussion in the context of prior work investigating the role of ITGA2 in OC. For example, Shield et al., 2007, showing a prominent role for ITGA2 in spheroid formation and disaggregation and proteolysis required for OC dissemination, and that function blocking antibodies abrogate these effects. In addition, a recent paper by Ma et al., 2020, focused on the role of ITGA2 in OC asking similar questions to those in this submission, with some notable similarities and differences in key findings.

15) The data in Figure 3 lack relevance to collagen (unless the cells secrete abundant collagen, which has not been investigated). Why are the proliferation and colony growth assays performed on plastic rather than in on collagen?

---

## [Author Response]

Essential revisions:1) Although ITGA5 is listed as non-significantly elevated in patient samples, it looks as if very few normal specimens were profiled for this integrin, compared to the others, for some reason.

We are not completely sure which figure should refer to this question, however, the total number of normal human omentum specimens profiled for ITGA5 is identical as ITGA2 as shown and described in the Figure 1E (n=19).

This is a particularly important issue to evaluate carefully, given several publications that have noted a specific and critical role for ITGA5 in promoting ovarian cancer metastasis (among others, PMID: 30710055, PMID: 23499550, PMID: 18381440), given the images provided in Figure 1E suggest the expression of this protein is particularly heterogeneous among tumors, and given the authors' data in Figure 5A actually suggests it is abundant. They also exclude ITGA5 from important subsequent analysis (e.g., database and direct cell line evaluation in Figure 2, 4, etc). This eliminates a critical benchmark; for example, the authors demonstrate using three cell lines in which ITGA5 is knocked out that binding to collagen is lost. If these three lines happen to be rare ones lacking ITGA5 expression, this could explain the dramatic phenotype. Information about ITGA5 should be added consistently.

We appreciate the important role of the fibronectin binder ITGA5 in ovarian cancer metastasis. It is well described in the current literature as pointed out by the reviewers. The original manuscript has already mentioned the involvement of ITGA5 in mediating cancer cell interaction to fibronectin as one possible disseminating process in peritoneal metastasis through an c-Met/FAK/Src dependent signaling (Gao et al., 2019; Iwanicki et al., 2011; Kenny et al., 2014) which can be found in the Introduction and Discussion. However, we do see the point raised by the reviewer and aimed to incorporate the ITGA5 more prominent in the context of our newly suggested ITGA2-collagen driven mechanism promoting ovarian cancer metastasis to the omentum. Accordingly, we evaluated the expression of ITGA5 consistently in the revised manuscript.

We agree with the reviewer on the heterogeneous and abundant expression of ITGA5 among tumor samples as seen in Figure 1E and 5A, respectively. We explain the difference noticed by the reviewer with the different experimental methodology applied as well as human specimens investigated. The Western blot analysis in bulk tumor tissue whole cell protein lysates (Figure 1E) shows the expression of ITGA5 independent of composition and cell types (e.g. stromal, immune, endothelial, and tumor cells). In contrast, the flow cytometry analysis in Figure 5A reflects the cell surface protein expression in the tumor spheroid isolated from patient-derived ascites. The high abundance of ITGA5 in those samples is also consistent with a previous finding demonstrating elevated ITGA5 expression in ovarian cancer spheroids as compared to the primary tumor for promoting cell adhesion and peritoneal metastasis (Gao et al., 2019). Upon request and to further strengthen our data, we have consistently added supplementary data for ITGA5 expression to the CCLE dataset as well as our cell line panel used throughout the manuscript (Figure 2B, Figure 2E, Figure 5—figure supplement 1C, and Figure 7H). The generated data support our initial findings that altered ITGA5 expression unlikely associated with ITGA2-mediated cancer cell-collagen binding.

Refer to the last question, the three *ΔITGA2* cell lines (SKOV3, OVCAR3, and IGROV1) used for cell-ECM binding assay showing completely lost of collagen binding is independent of ITGA5 expression.

2) In a survey of cell models, ITGA2 expression did not correlate with histotype or TP53 mutation status. This is curious, since serous ovarian cancers are the most aggressive and metastatic; one might expect to see higher levels in this cancer type. What is the author's explanation for this?

We thank the reviewer for pointing out that there is no correlation between *ITGA2* expression and serous histotype/ *TP53* mutational status in the CCLE dataset (Figure 2A and 2C). In consistency with the cell line survey mentioned by the reviewer, already provided ovarian cancer transcriptomic data GSE51088 (n=172) (Karlan et al., 2014) as well as our own cohort (n=200) on ITGA2 protein expression suggest an histotype-independent ITGA2 expression considering *TP53* mutation as dominating genetic alteration in serous ovarian cancer. An inconsistent pattern in this regards was observed for three additional publicly available datasets showing a significant upregulation of *ITGA2* expression in endometrioid and clear cell ovarian cancer in both (GSE73614; n=107) (Winterhoff et al., 2016) and GSE2109 (n=204). In contrast, no significant difference was observed for GSE9891 (n=285) (Tothill et al., 2008) (Figure 2—figure supplement 2A). The discrepancy among datasets may likely due to the different methodology applied, the biological heterogeneity of the ovarian tumor samples investigated, or the overall insufficient numbers of non-serous histotypes of ovarian cancer. Additionally, we also confirmed that the mutational status of *TP53* does not correlate with ITGA2 expression (Figure 2—figure supplement 2B-C). Taken together, having investigated five independent transcriptomic data sets on ITGA2 gene and protein expression without revealing a consistent pattern throughout the majority of the studies further support our initial data that ITGA2 cannot clearly be linked to current clinicopathological parameters or mutational status.

3) In Figure 5, the PIs use the ITGA2-specific inhibitor in cell lines overexpressing both ITGA2 and ITGA5 to demonstrate that inhibition of ITGA2 is sufficient to disrupt binding to collagen. However, they use this inhibitor at 40 μM. From the Tocris web page, "Potent α2β1 integrin inhibitor (IC50 values for the inhibition of human platelet adhesion to type I collagen are 12 and 715 nM for platelets under static conditions and under flow, respectively). Displays selectivity for α2β1 over αvβ3, α5β1, α6β1 and αIIbβ3 at concentrations exceeding 1000 nM." 40 μM is far in excess of these concentrations. The assay should be performed at appropriate concentrations; for instance, 1 μM. In addition, Tocris sells an α5 inhibitor, ATN161; this should be used in parallel assays, with the expectation it would not have an effect at concentrations where it is active for ITGA5 but not ITGA2.

We thank the reviewer for this valuable comment. As requested, we have tested a newly manufactured batch of α2β1 integrin inhibitor and repeated all inhibition assays using 1μM (Figure 5D-E). In brief, the inhibitor specifically blocks ITGA2-collagen interaction in a lower range of 0.2-1μM and is in line with data initially reported for 40 μM.

Additionally, we tested the suggested ITGA5 inhibitor ATN161, a noncompetitive inhibitor of the fibronectin PHSRN sequence. However, we could not observe any reduction in cell-adhesion to fibronectin even up to 100 μM treatment using experimental conditions as applied for ITGA2. This is consistent to previous studies showing that ATN-161 does not block integrin-dependent adhesion to ECM (fibronectin), but may inhibit integrin-dependent signaling as part of its mechanism of action (Plunkett and Mazar, 2002)

In order to demonstrate the ligand specificity of both ITGA2 and ITGA5, we additionally generated ITGA5-KO cells using CRISPR-*Cas9* in the IGROV1 and SKOV3 cell line (detailed information provided in the Materials and methods). Genetic loss of ITGA5 was confirmed by flow cytometry and Western blot analysis (Figure 5—figure supplement 2). The cell-ECM adhesion of *ΔITGA5* revealed the expected reduction to its well-known ligand, fibronectin, but only marginal change for binding to collagen. These results indicate that the potential cross-reactivity of cell surface ITGA5 to collagen is marginal and are in line with our response to reviewer’s comment 1 above.

4) Interestingly, from the data in Figure 5A show a partial although incomplete overlap between ITGA2 and ITGA5 expression. In Figure 5B, the authors show "a strong positive correlation between the percentage of ITGA2+ ATCs and the adhesion efficiency to collagen". A similar analysis should be done for ITGA5.

We re-analyzed the correlation of ITGA2 and ITGA5 expression with regards to cell-ECM adhesion efficiency for collagen I and fibronectin (Figure 5B). In brief, cell surface ITGA2 levels positively correlate with cancer cell adhesion to collagen I (R=0.6858, **p*=0.0286) but not correlate to fibronectin (R=-0.04, *p*=0.902). *Vice versa*, cell surface ITGA5 levels positively correlate only with fibronectin (R=0.6694, **p*=0.0342) but not collagen I. Taken together, we provide further evidence that *ΔITGA5* marginally impacts cell-collagen binding (please see response 3), and that ITGA2 and ITGA5 show independent receptor-ligands specificity.

5) The importance of FAK signaling in EOC is well-established, and targeting of FAK is already being explored clinically in these cancers (for instance, PMID: 31478830), reducing the novelty of the last part of the study. Hence, bolstering conviction of an activity specific for ITGA2 is important. FAK is activated by multiple integrins, and its activation in response to ITGA5 is well-established. It would be important to benchmark ITGA2 and ITGA5 in some of the experiments in Figure 7. For instance, do the cell lines analyzed by phosphoproteomic analysis express or not express ITGA5? Do the metastatic cells in the omentum with high ITGA2 also have ITGA5? Are there examples with high ITGA2 but not ITGA5? These seemed to be relatively rare based on data shown in Figure 5A. These data are extremely important for assessing the specificity of the signaling, and in understanding the relevant clinical populations likely to respond to FAK inhibitors.

We agree with the reviewer’s statement as it is indeed well known that multiple integrins triggered the activation of FAK signaling including ITGA5 (Harburger and Calderwood, 2009), and the oncogenic role for FAK activation has been shown to sustain tumor sphere proliferation and chemo-resistance in ovarian cancer (Diaz Osterman et al., 2019). Here, we could not exclude the involvement of ITGA5-dependent activation of FAK signaling. However, the selection of five ovarian cancer cell lines for gene editing represents the heterogeneity of ovarian cancer displaying differential expression of both integrins as shown in Figure 2E (see also comment 1). As suggested, we investigated additional cell lines (*ΔITGA2* in SKOV3 and OVCAR3) expressing low levels of endogenous ITGA5 using phosphoproteomic analysis (Please see further details in comment 13).

To further support our finding in addition to phosphoproteomic analysis, we also analyzed omental metastasis samples shown in Figure 7H of the original submission, here demonstrating that ITGA5 expression is abundant in almost every sample but less likely linked to activate FAK-dependent signaling. As a result, the significant fold change of identified phosphoproteins under collagen stimulation condition is most likely the causality of ITGA2 deletion.

6) It is actually interesting that expression of ITGA2 and ITGA5 seem to correlate (based on data provided). Do the authors have any thoughts as to whether this correlation is real, and if so, what is causing it?

We apologize that our statement is not clear that although the expression pattern of ITGA2 and ITGA5 in our ascites samples (N=20) seems partly overlap, but there is no significant correlation between the integrins with regards to expression in our cohort. Of note, it is widely known that to sustain the anchorage-independent growth and anoikis resistance, ascites-derived tumor spheroids undergoing epithelial-mesenchymal transition (EMT) and upregulated integrin expression to facilitate cell-ECM adhesion and subsequent tumor implantation (Ma et al., 2020). Such a unique ascites microenvironment may explain the positive upregulation of both integrins on tumor spheroids, however, the molecular signatures of ascites-derived tumor cells and their primary and metastatic tumor counterpart likely being significantly different (Gao et al., 2019). We also examined the correlation of the collagen binder ITGA2 to fibronectin binder ITGA5 in the CCLE dataset. We now include the correlation scatter plot in Figure 5—figure supplement 1 and show that there is no significant correlation between ITGA2 and ITGA5 expression in CCLE (ovarian cells *p*=0.46, all cancer cells *p*=0.057) dataset.

7) The data in Figure 5-7 are the most exciting. Are the metastatic foci rich in collagen? Does FAK inhibition have an effect on this? FAK is a key regulator of anoikis but also functions downstream of integrin to regulate adhesion turnover etc. Is the FAKi effect (Figure 7G-H) limited only to inhibiting sphere growth or does it also influence the mesothelium clearance/collagen interaction? The authors should clarify their description and interpretation of the elevated expression of collagens in omentum bearing metastases. Certainly, the fact that the omentum changes expression of collagens over time is well known, as is the relationship of this to onset of aging-related pathologies (one recent review – PMID: 29996539). Some of these changes may "prime the soil" for seeding by ovarian cancers expressing cognate integrins. However, the analysis of phosphoproteomic and TCGA data at the end of the paper seems to suggest the collagens are being produced by the tumor cells themselves – which would lead to a different model.

We thank the reviewer for this very positive and enthusiastic feedback. The initial thought of our study was based on the assumption that the ECM composition in the omentum contributes to early onset of metastasis and thus, serves as pre-metastatic niche. Beside the established ITGA5-fibronectin interaction in ovarian cancer, we noticed that the normal omentum is rich in collagen (Figure 1C) and particularly type I collagen (triple helix of COL1A1 and COL1A2; (Figure 1—figure supplement 1A). Therefore, we have analyzed the collagen expression in normal and metastatic omentum by immunohistochemistry and found that the metastatic foci is also rich in collagen (Figure 1—figure supplement 1B). In line with the proteomic analysis already shown in Figure 1A, that COL1A1 is significantly upregulated in omental metastasis samples. A recent study has well-described that the upregulation of COL1A1, COL11A1, VCAN, FN1, COMP, and CTSB, is associated with cancer progression, poor prognosis, and malignant cell invasion in ovarian cancer (Pearce et al., 2018). Interestingly, the authors also reported that normal omental ECM stained highly positive and evenly distributed COL1A1, but malignant omentum showing strong but uneven staining likely due to the expression of matrix remodeling proteases.

In regard to the last question, *th*e analysis of phosphoproteomic and TCGA data in Figure 7—figure supplement 1B is mainly to provide evidence that ITGA2 expression positively correlates and various ECM proteins including collagens. Of note, we should be aware that the transcriptomic analysis of tumor is based on bulk RNA-sequencing and therefore not necessary being produce by tumor cells itself. Additionally, to address this interesting thought on the potential impact of self-secreted collagen by cancer cells, we analyzed the mRNA expression of collagens in the CCLE ovarian cancer cell lines dataset. Ovarian cancer cell lines express predominantly COL4A1, COL4A2, and COL18A1 whereas human fibroblasts are well-known to produce high abundant of COL1A1 and COL1A2 give rise to type I collagen (Figure 1—figure supplement 2). The origin of high COL1A1, COL1A2, COL3A1 expression identified in the normal omentum and omental metastases positively correlate with fibroblast-specific markers including fibroblast activation protein 1α (FAP), and α-smooth muscle actin (ACTA2) described in Figure 1—figure supplement 1C of the original manuscript. Although we could not completely exclude the impact of self-secreted collagens by ovarian cancer cells, our in vitro colony formation and proliferation assay (please see our response to comment 15 for more details) revealed that cancer cell-secreted COL4 and COL18 may not be the primary ligands for ITGA2-dependent cell growth.

8) Although ITGA2 has not been as heavily investigated as other integrins for a role in EOC, one study found it to be upregulated in ovarian cancer (PMID: 31443152), and a more recent one showed it contributed to paclitaxel resistance in ovarian cancer by activating AKT/paclitaxel (PMID: 32202508). The discussion should discuss these works in greater detail; in particular, it would be useful to place the current work in the context of AKT pathway activation, given this is a major regulator of anoikis, and engages in cross-talk with FAK, SRC, and ERK.

We have now addressed the recent critical work about ITGA2 in ovarian cancer in the discussion according to the reviewer’s suggestion (please see response to comment 14). Briefly, Ma *et al.*, demonstrated the AKT/Foxo1 signaling axis confer paclitaxel resistance and cell survival in three ovarian cancer cell lines (OVCAR3, A2780, and SKOV3) in vitro and xenograft model. They showed that phosphorylation of AKT S473 is increased in an ITGA2-dependent manner. Although we did not identify the downregulation of pS473 in ITGA2 knockout cells using a phosphoproteomic discovery approach, the phosphorylation of AKT S124 and S129 are significantly reduced in the ITGA2-collagen signaling axis in both EOC cell lines (Figure 7—figure supplement 1D) and patient samples (Figure 7H). Both phosphorylation sites were required for elevated AKT1 activity (Di Maira et al., 2005), suggesting that AKT activation is also engaging in cross-talk with ITGA2-dependent FAK signaling.

9) Authors state that ΔITG2A cells "maintained adhesion to laminin and fibronectin (Figure 4, A-B)", but the data showed variability across cell lines, suggesting this was not generalizable. While true for SKOV3 cells, other cell lines showed significant differences in one or the other substrate. This variability in effect on FN or LN binding is also observed in experiments utilizing the ITGA2 inhibitor TC-I-15 (shown in Figure 5). The text describing the findings in the context of FN and LN should be modified.

We apologize for the insufficient description of the data obtained for cell binding assays to FN and LN. We made the correction to the corresponding text in Figure 4A-B.

“Notably, the consistently strong cell adhesion efficiency to collagen type I, III, and IV in parental cell lines significantly diminished upon ITGA2 deletion. However, cell adhesion to laminin and fibronectin appeared in a cell line-dependent manner in ΔITGA2 cell lines which may be due to the redundancy of multiple integrin subunits. (Figure 4, A-B).”

10) Although the authors tested the effects of TC-I-15 on AS20 cells and showed no significant change, use of these cells only does not fully address the specificity of TC-I-15 for ITGA2 as ITGA5 protein levels in these cells are mid-range. Testing of cells that exhibit ITGA2 low ITGA5 high would further support the selectivity of drug effect for ITGA2.

As requested, we selected three independent group of ATCs that exhibits ITGA2^low^/ITGA5^high^ (AS14, AS15, AS17), ITGA2^high^/ITGA5 ^high^ (AS4, AS11, AS18) and ITGA2^low^/ITGA5^low^ (AS13, AS19, AS20) for subsequent cell-ECM adhesion analysis in the presence of TC-I-15. Related data are shown in Figure 5. In line with our previous finding, re-analysis of ATCs (n=9) adhesion suggested that TC-I-15 treatment can specifically block ITGA2 interaction to collagen with marginal difference to fibronectin and importantly, independent of ITGA5 expression (Figure 5D). However, whether the discrepancy of cell-collagen adhesion efficiency between ITGA2^low^/ITGA5^high^ and ITGA2^low^/ITGA5^low^ is due to the redundancy of other collagen-binding integrins or not (e.g. ITGA1, ITGA10 or ITGA11), requires further investigation. Furthermore, the ligand-binding specificity is also supported by our newly established *ΔITGA5* cells showing decreased adhesion to fibronectin but not to the collagens as discussed above (see response 3, Figure 5—figure supplement 2).

11) The finding that TC-I-15 has no effect on spheroid formation seems at odds with the prediction that ITGA2 is a key contributor to – if not essential for – sphere formation and escape from detachment-induced anoikis as shown in Figure 3 D and E. The authors show OE of ITGA2 results in increased anchorage independent colony formation. Was the effect of deletion of ITGA2 on anchorage independent colony size and number tested (WT vs ΔITG2A cells)?

We are also aware of the reviewer’s concern about the cancer sphere formation. The initially submitted manuscript showed that overexpression of ITGA2 increased colony forming ability in BG1 and A2780 cells (Figure 4—figure supplement 1E). However, other cell lines (SKOV3, OVCAR3, OAW42, and ES2) did not form rigorous spheroids using the anchorage-independent soft agar assay. Thus, we could not generalize the effect of deletion of ITGA2 for those cell lines. Nevertheless, we additionally performed the anchorage-independent spheroid formation for IGROV1 cells in the ITGA2 knockout background in order to acknowledge reviewers comment. In line with previous findings, we found significantly reduced colony formation in *ΔITGA2* cells (both diameter and number of spheroids) as compared to the parental IGROV1 cells. Data obtained have been added to Figure 4—figure supplement 1F.

In regard to the effect of TC-I-15 on spheroid formation, the inhibitor itself does not involve in reducing transcriptional and translational activity of ITGA2 expression, however acting as an allosteric inhibitor blocking collagen-ITGA2 engagement site and subsequent signaling (Miller et al., 2009). Therefore, other ITGA2 binding ligands may remain actively involved in ITGA2-mediated signaling in spheroid formation. Besides canonical integrin signaling, ligand-independent integrin signaling has also been reported through the anchoring receptor-mediated cell adhesion to transmit a mechanical stimulus to the integrin that signals independently of ECM binding (Ferraris et al., 2014)

12) in vivo analysis of SKOV3 WT and ΔITGA2 cells injected in nude mice showed significant reduction of total tumor implants in the ΔITGA2 xenografts. In Figure 3D, the authors show that SKOV3 ΔITGA2 cells grown on non-adherent plates exhibited a high degree of cell death (46.6%) as compared to SKOV3 control cells (12.9%). Was the potential difference in viability of the injected cells addressed to exclude the possibility that the reduction in tumor growth was due to significantly reduced viability of ΔITGA2 resulting in fewer cells to capable of growth? This is an essential point to clarify before one can conclude that ITGA2 is promoting dissemination and invasion of collagen rich omentum.

We fully agree with the reviewer and appreciate for pointing this out. Indeed, the *ΔITGA2* cells showed significantly increased anoikis compared to parental cells 3 days after cultivation in non-adherent condition (Figure 3D). We performed additional experiments in reply to comment 15 below and observed a significant cell proliferation benefit on collagen-coated plates in parental cell lines as compared to *ΔITGA2* cells (Figure 3A-B). These results suggest that ITGA2-mediated adhesion of ovarian cancer cells to the ECM is critical for cell survival. The elevated cell death only occurs while cells cultivated in a non-adherent condition, however, this is unlikely to be in the case of mouse abdominal cavity in which the peritoneum or omentum are covered by a mono-layer of mesothelium and richness of extracellular matrix proteins. This is also supported by a recent study shown by Gao et al. demonstrating that intraperitoneal injection of SKOV3 cells and cancer-associated fibroblast (CAFs) were able to instantly adhere to the metastatic tropism of omentum, peritoneum, and mesentery 4 hour after initial injection of tumor cells. Further spreading and development of tumor nodules was also pronounced in 1-2 weeks. Additionally, we did not identify a significant reduction of the *ΔITGA2*-derived tumor implants in organ such as peritoneum, intestine, kidney, stomach and lung (Figure 6H). Overall, we are confident the reduction of tumor growth in the omentum is not primarily due to the reduced viability in *ΔITGA2* cells.

13) Phosphoproteomic analysis led the authors to conclude that ITGA2-collagen interaction fosters cancer cell survival, cell migration and anoikis resistance through the FAK/MAPK signaling axis. The only cell line mentioned in this analysis is IGROV-1. The implications of the findings would be substantially strengthened by analysis of additional cell lines with comparative analysis across lines to substantiate key findings. Functional assays would lend further support to the conclusion that FAK/MAPK signaling axis is the primary mechanism.

We agree with the reviewer’s statement that the implication would be substantially strengthened by analyzing additional cell lines. We performed phosphoproteomic analysis in two additional cell lines which are both ITGA2^+^ but ITGA5^Low^ (SKOV3 and OVCAR3, please also refer to comment 8). Despite less phosphoproteins identified in those two additional cell lines, we identified 69 down-regulated phosphoproteins shared across all three *∆ITGA2* cell lines (Figure 7A-B, and Supplementary file 1B). Next, we applied the gene ontological (GO) analysis to classify the biological process regulated by ITGA2-collagen signaling using Gene Ontology Database (DOI: 10.5281/zenodo.3954044 Released 2020-07-16). We found that the top 7 cell components enriched in the genes regulated by ITGA2, were related to cell organization, polarity and GTPase activity, which was further supported by GSEA analysis revealing that MAPK1/3 activation and cell junction organization being significantly regulated by ITGA2 axis (Figure 7C-D). Importantly, down-regulation of the T204 phosphorylation on MAPK3 and Y577, T700, S843 phosphorylation on FAK were also present in the additional two cell lines (Figure 7—figure supplement 1C) as well as in omental metastasis tissues (Figure 7H). Data of phosphoproteomic analysis were provided in Figure 7—source data 1.

To further address reviewer’s concern about supporting FAK and MAPK as the primary mediators of the ITGA2-collagen dependent signaling, we performed cell adhesion and mesothelial clearance assay in the presence of ITGA2, FAK and MAPK(ERK1/2) inhibitors, respectively. Interestingly, only TC-I-15 treatment blocks cell adhesion to collagen (Figure 7—figure supplement 3E). Blockade of either FAK or MAPK signaling axis showed no direct impact of cell-adhesion to collagen I in vitro. It is reasonable that the FAK/MAPK-dependent cell survival is the downstream of integrin receptor signaling, therefore ITGA2-mediated initial cell adhesion is marginally affected. On the other hand, blocking ITGA2 or downstream FAK/MAPK signaling pathway significantly reduced mesothelial clearance activity of ovarian cancer cells, supporting the functional involvement of ITGA2 and FAK/MAPK axis in ovarian cancer metastasis model (Figure 7—figure supplement 3F). Furthermore, in line with our previous finding showing that Defactinib treatment inhibiting p-FAK^Y397^, p-FAK^Y576/577^ and the downstream effector cyclin D1 expression (Figure 7F), two cell lines share also a similar trend with supporting the oncogenic role of FAK/MAPK signaling involved in cell survival, anoikis resistance and ITGA2-dependent cancer metastasis (Figure 7—figure supplement 2).

14) The findings of this study should include deeper discussion in the context of prior work investigating the role of ITGA2 in OC. For example, Shield et al., 2007, showing a prominent role for ITGA2 in spheroid formation and disaggregation and proteolysis required for OC dissemination, and that function blocking antibodies abrogate these effects. In addition, a recent paper by Ma et al., 2020, focused on the role of ITGA2 in OC asking similar questions to those in this submission, with some notable similarities and differences in key findings.

In order to acknowledge the reviewers’ comment on an extended discussion about recent research published by Shield et al., and Ma et al. focusing on the role of ITGA2 in ovarian cancer, as well as different types of cancers, we have modified the Discussion accordingly as shown below:

“The role of ITGA2 remains controversial in different cancer types with elevated ITGA2 expression reported to exacerbate experimental metastasis in melanoma, gastric, and colon cancer (Baronas-Lowell et al., 2004; Bartolome et al., 2014; Matsuoka et al., 2000){Yoshimura, 2009 #1}.{Yoshimura, 2009 #1}{Yoshimura, 2009 #1}{Yoshimura, 2009 #1} […] As a result, a comprehensive understanding of integrin-ECM interaction, a crucial step for cancer cell dissemination, is uncertain but of immense importance for development of potential therapeutics targeting peritoneal metastasis.”

15) The data in Figure 3 lack relevance to collagen (unless the cells secrete abundant collagen, which has not been investigated). Why are the proliferation and colony growth assays performed on plastic rather than in on collagen?

We are thankful to reviewer’s great suggestion. Intrigued by the comment we have analyzed the cell proliferation in the presence of either collagen or fibronectin in our entire cell line panel. In brief, the cell proliferation was examined in the presence of type I collagen and fibronectin using MTT and colony formation assay. As the reviewer’s might expect, significantly reduced cell proliferation was observed in *∆ITGA2* cells only on collagen coated wells. In contrast, neither classical plastic nor fibronectin coated plates showed such a difference (Figure 3A-B and Figure 3—figure supplement 1). Besides providing further evidence on the specificity of ITGA2-mediated binding to collagen, these results also support that the collagen-ITGA2 dependent axis might be an important mechanism in ovarian cancer cell proliferation and tumorigenicity.